# Cluster-Dags as Powerful Background Knowledge For Causal Discovery

**Jan Marco Ruiz de Vargas**                                      *janmarco.ruiz@tum.de*
*Technical University Munich*

**Kirtan Padh**                                                    *kirtan.padh@tum.de*
*Technical University Munich*
*Helmholtz Munich*
*Munich Center for Machine Learning (MCML)*

**Niki Kilbertus**                                                 *niki.kilbertus@tum.de*
*Technical University Munich*
*Helmholtz Munich*
*Munich Center for Machine Learning (MCML)*

**Reviewed on OpenReview:** *https://openreview.net/forum?id=gSSmvVDKxB&referrer*

## Abstract

Finding cause-effect relationships is of key importance in science. Causal discovery aims to recover a graph from data that succinctly describes these cause-effect relationships. However, current methods face several challenges, especially when dealing with high-dimensional data and complex dependencies. Incorporating prior knowledge about the system can aid causal discovery. In this work, we leverage Cluster-DAGs as a prior knowledge framework to warm-start causal discovery. We show that Cluster-DAGs offer greater flexibility than existing approaches based on tiered background knowledge and introduce two modified constraint-based algorithms, Cluster-PC and Cluster-FCI, for causal discovery in the fully and partially observed setting, respectively. Empirical evaluation on simulated data demonstrates that Cluster-PC and Cluster-FCI outperform their respective baselines without prior knowledge.

## 1 Introduction

Understanding causal relationships is essential for scientific inquiry and reasoning about the world. Researchers have always leveraged active experimentation and interventions to uncover causal mechanisms. But in many cases, such experiments are impractical or ethically off-limits—for instance, it would be unethical to expose communities to varying levels of environmental pollution to study their effect on health. The challenges of experimentation, along with the data explosion in recent decades, have led to continued methodological advances in causal discovery from observational data (Guo et al., 2020; Malinsky & Danks, 2018; Peters et al., 2017).

One common approach to represent causal assumptions is the Structural Causal Model (SCM) framework (Pearl, 2009a; Peters et al., 2017), which encodes these relationships in Directed Acyclic Graphs (DAGs), with edges representing the "directed flow of causal influence" between variables. One landmark result in causal discovery is that, without further assumptions, purely from observational data one can only identify the Markov equivalence class—a collection of graphs that are all consistent with the observed data. In large graphs, this ambiguity can make it difficult to pinpoint precise causal insights (Spirtes et al., 2000; 1995). Moreover, as the number of variables increases, the space of DAGs grows exponentially, posing serious scalability issues for many causal discovery algorithms (Ganian et al., 2024).

One way to help address these challenges is by incorporating background knowledge—partial information about the graph structure that can narrow the search space and guide causal inference. A recent method for encoding such partial knowledge is via Cluster-DAGs (C-DAG) (Anand et al., 2023), which organize variables into clusters and assume that possible causal relationships between these clusters are known, but the causal structure within the clusters as well as the precise connections of individual variables across clusters is still unknown. By using structural information at this higher level, C-DAGs offer a way to manage complexity in high-dimensional settings. C-DAGs were originally developed for direct use in causal inference, reasoning about the strength of cause-effect relationships using the C-DAG directly (Anand et al., 2023).

We leverage C-DAGs as background knowledge to improve constraint-based causal discovery methods using the prior knowledge *during* discovery process itself. We also compare it to the related tiered background knowledge proposed by (Andrews et al., 2020). Our contributions include:

- We show that tiered background knowledge (TBK) (Andrews et al., 2020) can be represented as a C-DAG, but not vice-versa—C-DAGs capture strictly more flexible types of background knowledge. In addition, C-DAGs can accommodate latent confounding between clusters, whereas TBK has to assume it non-existent, see Section 2.1.
- We formalize the constraints a C-DAG puts on a corresponding DAG by formulating the C-DAG restrictions as a boolean combination of pairwise constraints, see Appendix C for details. This could be interesting for future theoretic research in causal discovery with background knowledge.
- C-DAGs allow warm-starting of constraint-based causal discovery algorithms, namely PC (Spirtes et al., 2000) and FCI (Zhang, 2008b) by pruning and orienting edges before running the discovery algorithm. This results in fewer CI tests needing to be performed. We introduce the Cluster-PC and Cluster-FCI formally and show their soundness (C-PC and C-FCI) and completeness (for C-PC only, C-FCI is not be complete by design).
- Our simulations show that Cluster-PC and Cluster-FCI outperform the non-background knowledge baselines, see Sections 3 and 4. Cluster-FCI also outperforms FCITiers on C-ADMGs (allowing bidirected edges between clusters), while being close to FCITiers on C-DAGs.

C-DAGs naturally arise in many scientific domains where prior knowledge organizes variables into functional groups. For example, in systems biology, genes belong to known pathways (KEGG, Reactome) with established directional relationships between pathways (Dugourd et al., 2021b). In climate science, variables cluster by physical processes (ocean, atmosphere, cryosphere) with causal pathways identified through process-based understanding (Runge et al., 2019a). In social epidemiology, the 'web of causation' organizes social determinants into domains with complex causal interdependencies (Korvink et al., 2025a). In each case, the cluster-level causal structure contains v-structures ($C_1 \rightarrow C_2 \leftarrow C_3$) that TBK cannot represent, making C-DAGs strictly more appropriate.

## 1.1 Related work

**Background Knowledge in Causal Discovery.** There are two main ways in which background knowledge can be described in graphical structures: groupwise background knowledge and pairwise background knowledge. Pairwise background knowledge (Fang et al., 2025; Meek, 1995) restricts the relationship between pairs of variables, i.e., requiring or ruling out directed edges or ancestral relationships. In contrast, groupwise background knowledge (Andrews et al., 2020; Brouillard et al., 2022; Anand et al., 2023) organizes variables into different groups and then restricts the edges between these groups, without committing to any pairwise constraint directly. Previous work on background knowledge includes guiding score-based methods like KGS (Hasan & Gani, 2024), which can use prior knowledge on the absence/presence of an (un)directed edge, $A^*$-based methods (Kleinegesse et al., 2022), using absence/ presence of directed edges and tiers (non-ancestral constraints) and NOTEARS (Chowdhury et al., 2023), using absence/ presence of directed edges. Most of these methods focus on pairwise background knowledge. For constraint-based methods, except for (Andrews et al., 2020), background knowledge is usually used to orient additional edges after receiving the CPDAG (Brouillard et al., 2022; Bang & Didelez, 2023) from the PC algorithm (Spirtes et al., 2000). Fang et al. (2025) study the representation of causal background knowledge using pairwise

background knowledge. Pairwise and groupwise background knowledge can be combined, although to the best of our knowledge, not much research has gone in this direction yet.

**Causal graphs over groups of variables and causal abstractions.** Recently, different aspects of causal diagrams over groups of variables have been discussed. Parviainen & Kaski (2017) study learning groups of variables in Bayesian networks. Wahl et al. (2023) discuss two methods for inferring causal relationships between two groups of variables. Melnychuk et al. (2024) use C-DAGs to group confounders as a cluster. Wahl et al. (2024) extend the theoretical framework of Anand et al. (2023), discussing the relationships between micro and group level graphs. C-DAGs and other variable aggregation methods have increasingly been used to model causal systems (Ma et al., 2025; Plecko & Bareinboim, 2024; Raghavan & Bareinboim, 2025; Assaad et al., 2024; Xia & Bareinboim, 2025; Tabell et al., 2025). Such an aggregate model can then, with our approach, be used to inform causal discovery of the more granular DAG. Group graph variants have also increasingly become targets for causal discovery (Niu et al., 2022; Ninad et al., 2025; Göbler et al., 2025; Anand et al., 2025). Xia & Bareinboim (2024) use neural causal models to learn neural causal abstractions by clustering variables and their domains, while Massidda et al. (2024) create a variant of LiNGAM (Shimizu, 2014), Abs-LiNGAM, to learn causal abstractions. Li et al. (2025); Yvernes et al. (2025d) investigate the identifiability of causal abstractions, Schooltink & Zennaro (2025) bridge graphical and functional causal abstractions and Massidda et al. (2025) study causal sufficiency for causal abstractions. While much work focuses on learning the aggregated group graph itself, our approach uniquely leverages the C-DAG as assumed prior knowledge to perform causal discovery that resolves the relationships within the underlying micro-variables.

**Summary Graphs and Identifiability.** The CaGreS algorithm by Zeng et al. (2025) summarizes DAGs via node contractions and creates summary causal graphs (SCGs) with preserved utility for causal inference. Ferreira & Assaad (2025b) extend on Anand et al. (2023) by analyzing theoretical properties of SCGs, which are similar to C-DAGs, but also allow for cycles. Yvernes et al. (2025a); Assaad et al. (2024); Assaad (2025); Yvernes et al. (2025b;c); Ferreira & Assaad (2025a) investigate identifiability in SCGs, while Ferreira & Assaad (2025c) do so for cluster directed mixed graphs, a related concept. Transit clusters (Tikka et al., 2023), while similar, are specifically designed to cluster variables while preserving identifiability properties. Zhu et al. (2024) show that interventions in aggregation of variables (e.g., a surjective but non-injective function of cluster variables) are no longer well-defined. We build on these properties developed for C-DAGs to guide a constraint-based search for the detailed underlying DAG.

**Applications.** On the application side, C-DAGs have also shown some promise for improving causal inference tasks. Ribeiro et al. (2025) apply causal inference to assess malaria risk, and they mention that causal discovery at the cluster level improves interpretability. Anand & Hripcsak (2025) apply C-DAGs to determine causal effects in medicine. We discuss how C-DAGs occur naturally in various other application fields in Appendix E.

**Research gap.** Despite increasing interest in causally modeling groups of variables, exploiting often easily available groupwise background knowledge for causal discovery of the detailed graph remains underexplored. Groupwise background knowledge flexibly accommodates prior knowledge on varying levels of detail for different subsets of the variables, often rendering it much more realistic in practice. Some groups of non-ancestrality constraints (e.g., today can not causally influence yesterday) is much more readily available and justified than individual required/forbidden edges or highly granular assertions. C-DAGs encode such groupwise knowledge flexibly and in a visually interpretable, intuitive way, making them a valuable tool for applied researchers and users across domains.

## 1.2 Preliminaries on causality and constraint-based causal discovery

We briefly introduce the most relevant preliminaries for structural causal models (SCM) and causal discovery. For more details, we refer the reader to (Spirtes et al., 2000; Pearl, 2009b; Peters et al., 2017). Throughout, we assume a fixed set of random variables $X_1, \ldots, X_n$, which also serve as the node set of graphs $V = \{X_1, \ldots, X_n\}$.

A DAG (directed acyclic graph) $G = (V, E)$ is a directed graph over nodes $V$ with edges $E$ without directed cycles. If $(X \to Y) \in E$ in a DAG (for $X, Y \in V$), we write $X \in pa_Y$, $Y \in ch_X$ (parents, children,

respectively). The neighbors of $X$ are: $nb_X := ch_X \cup pa_X$. A superscript $G$ like $an_X^G$ indicates we are referring to the ancestors of $X$ in graph $G$. If there exists a directed path from $X$ to $Y$, i.e., $X \to \ldots \to Y$, then $X \in an_Y$, $Y \in de_X$ (ancestors, descendants respectively). When we find $X \to Y \leftarrow Z$ along a path in $G$, we call $Y$ a collider on the path; if $X, Z$ are additionally not adjacent in the DAG, we call the triple $\{X, Y, Z\} \subset V$ an unshielded collider or v-structure. For causal discovery, we will also consider acyclic graphs that can contain both directed and undirected edges (denoted by $X - Y$). This will often be interpreted as the direction of the edge not yet being specified or unknown.

A structural causal model over variables in $V$ entails both a probability distribution $P(X_1, \ldots, X_n)$, the observational distribution, and a DAG $G = (V, E)$. In fully observed SCMs, the observational distribution and the implied DAG are related by the Markov property: conditional independence statements about $P(X_1, \ldots, X_n)$ are implied by so-called d-separations in $G$. A path $X_i, \ldots, X_j$ in $G$ is called blocked by some set $S \subset V \setminus \{X_i, X_j\}$ if every non-collider on the path is in $S$ and every collider and their descendants are not in $S$. If all paths between $X_i$ and $X_j$ are blocked by $S$ in $G$, we say that $S$ d-separates $X_i$ and $X_j$ in $G$, denoted by $X_i \perp\!\!\!\perp_G X_j \mid S$. Shortly, $P(X_1, \ldots, X_n)$ satisfies the Markov property w.r.t. $G$ if $X_i \perp\!\!\!\perp_G X_j \mid S \Rightarrow X_i \perp\!\!\!\perp X_j \mid S$ in $P(X_1, \ldots, X_n)$.

For a one-to-one correspondence between d-separations in a graph $G = (V, E)$ and conditional independencies in a distribution $P(X_1, \ldots, X_n)$ to hold also requires the converse implication—called faithfulness. Faithfulness is not generally satisfied for the observational distribution and graph entailed by an SCM, but often assumed to hold in practice. The subtleties of the faithfulness assumption and its violations have been studied extensively in the causal discovery literature (Zhang & Spirtes, 2002; Ramsey et al., 2006; Uhler et al., 2013; Marx et al., 2021).

Under the assumptions of causal sufficiency—there are no unobserved common causes of any variables in $X$—and faithfulness, d-separations and conditional independencies are in one-to-one correspondence enabling constraint based causal discovery: by testing all possible conditional independencies in $P(X_1, \ldots, X_n)$ one can derive all d-separations that hold in $G$. It turns out that the set of DAGs that satisfies a given set of d-separations, called the Markov equivalence class, can be characterized as follows: Each DAG in the Markov equivalence class as the same skeleton (the same set of adjacencies) and the same set of v-structures (Pearl, 2009a; Zhang, 2008a). The Markov equivalence class can be represented by a partially directed graph (some directed, some undirected edges). If causal sufficiency is not assumed, the situation becomes more complicated and one can only infer more general types of graphs by testing conditional independencies. The two landmark algorithms for constraint based causal discovery with and without causal sufficiency are the PC and FCI algorithm, respectively. We will first focus on leveraging prior knowledge for a PC type algorithm in the fully observed case, before covering partial observations and an extension of FCI in Section 3.2.

## 2 Cluster-DAGs

Cluster-DAGs (C-DAGs) were introduced by Anand et al. (2023) as a way to perform causal inference when one can not specify the entire DAG, but has enough information to organize groups of variables into a DAG. C-DAGs aggregate variables into clusters and then define macro-relationships between these clusters. An example can be seen in Fig. 1.

No assumptions or restrictions are imposed on connections between variables within the same cluster. A directed edge between clusters $C_1 \to C_2$ means that for any vertices $X \in C_1, Y \in C_2$ there is either no edge between them or it is oriented as $X \to Y$. If there is no arrow between $C_1, C_2$, no nodes $X \in C_1, Y \in C_2$ are adjacent. Formally, a compatible C-DAG is defined as follows.

**Definition 1** (**C-DAG**, (Anand et al., 2023))**.** *Given an ADMG $G = (V, E)$ (a graph with directed and bidirected edges as in Definition 13) and a partition $C = C_1, \ldots, C_r$ of $V$ (i.e., $C_i \cap C_j = \emptyset$ for all $i \neq j$ and $V = \bigcup_{i=1}^r C_i$), construct a graph $G_C = (C, E_C)$ over $C$ with a set of edges $E_C$ defined as follows:*

*(i) An edge $C_i \to C_j$ is in $E_C$ if there exist $X \in C_i, Y \in C_j$ such that $(X \to Y) \in E$.*

*(ii) A bidirected edge $C_i \leftrightarrow C_j$ is in $E_C$ if there exist $X \in C_i, Y \in C_j$ such that $(X \leftrightarrow Y) \in E$.*

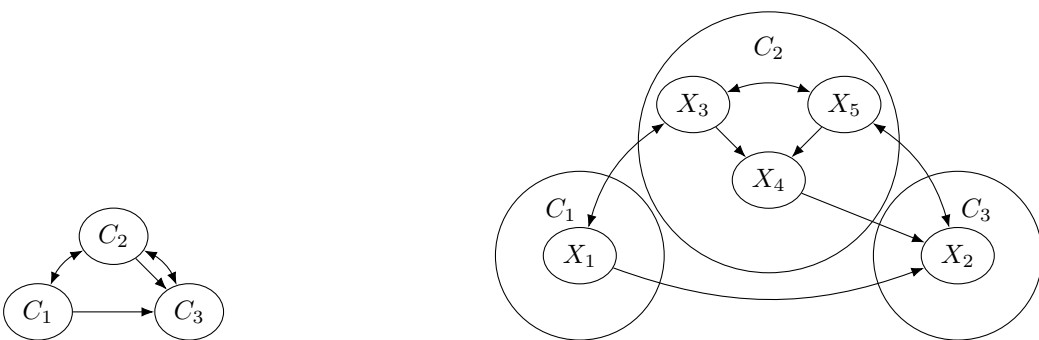

Figure 1: An example of a C-DAG $G_C$ and a compatible DAG $G$. **Left:** A C-DAG $G_C$ over three clusters. **Right:** A graph $G$ compatible with the C-DAG $G_C$.

*If the graph $G_C$ contains no directed cycles, $C$ is called an* admissible partition *of $V$ and $G_C$ is called a* C-DAG compatible with $G$. *Any ADMG $G$ that has $G_C$ as a compatible C-DAG is in turn called* compatible with $G_C$.

Many different DAGs may be compatible with the same C-DAG and vice versa, so C-DAGs form an equivalence class of DAGs. Anand et al. (2023) show that causal effects are still identifiable from the C-DAG $G_C$ if they are identifiable in *every* DAG $G$ compatible with $G_C$. Another important result is the soundness and completeness of d-separation in C-DAGs: any d-separation in the C-DAG also holds in any compatible DAG.

**Theorem 1** (**Soundness and completeness of d-separation in C-DAGs**, (Anand et al., 2023)). *In a C-DAG $G_C$, let $C_i, C_j, C_k \subset C$ be sets of clusters. If $C_i$ and $C_j$ are d-separated by $C_k$ in $G_C$ (see Definition 14), then in any ADMG $G$ (see Definition 13) compatible with $G_C$, $C_i$ and $C_j$ are d-separated by $C_k$ in $G$, that is*

$$C_i \perp\!\!\!\perp_{G_C} C_j \mid C_k \quad \implies \quad C_i \perp\!\!\!\perp_G C_j \mid C_k. \tag{1}$$

*If $C_i$ and $C_j$ are not d-separated by $C_k$ in $G_C$, then there exists an ADMG $G$ compatible with $G_C$ where $C_i$ and $C_j$ are not d-separated by $C_k$ in $G$.*

Based on these results, Anand et al. (2023) then develop an ID algorithm that is sound an complete for identifying causal effects in C-DAGs. In this work, we leverage implications of Theorem 1 for *causal discovery* instead.

In constraint-based causal discovery (Spirtes et al., 2000; Zhang, 2008b), one typically assumes faithfulness and removes edges from an initially fully connected graph whenever a conditional independence is found between any pair of vertices. The key idea is that Theorem 1 suggests that whenever $X \in C_i, Y \in C_j$ and $C_i \perp\!\!\!\perp_{G_C} C_j \mid S$, then in any $G$ compatible with $G_C$ we have $X \perp\!\!\!\perp_G Y \mid S$ (slightly abusing notation in that $S$ is a set of clusters or a union over them, respectively) and thus $X, Y$ cannot be adjacent. Hence, C-DAGs allow us to aggressively prune a fully connected graph to warm-start constraint-based causal discovery on the micro-variables. Besides this warm start that potentially saves many conditional independence tests, the C-DAG structure can also be used to further speed up the remaining discovery process, which we discuss in Section 3.

## 2.1 Comparison to tiered background knowledge

Before developing our full causal discovery algorithms with C-DAGs, we provide a thorough comparison with existing forms of groupwise background knowledge—tiered background knowledge (TBK). Like C-DAGs, TBK groups variables, where groups are called tiers. Unlike C-DAGs, tiers impose a directed, chronological ordering between groups of variables.

**Definition 2** (**Tiered background knowledge,** (Andrews et al., 2020)). *A MAG (maximal ancestral graph, Definitions 17 and 18) satisfies tiered background knowledge if the variables can be partitioned into*

$n > 1$ *disjoint subsets (tiers)* $T = \{T_1, \ldots, T_m\}$ *and for all* $A \in T_i$ *and* $B \in T_j$ *with* $1 \le i < j \le m$ *either (i)* $A$ *is an ancestor of* $B$ *or (ii)* $A$ *and* $B$ *are not adjacent.*

Similar to C-DAGs, TBK enforces the orientations of certain edges due to the macro-constraints. While causal discovery algorithms for TBK like FCI-Tiers (Andrews et al., 2020) have been developed, C-DAGs are strictly more flexible than TBK:

(i) TBK cannot represent settings like the C-DAG $C_1 \to C_3 \leftarrow C_2$. TBK would have to put either $C_1$ or $C_2$ first in the tier ordering, which would only restrict the orientation of edges between $C_1, C_2$, but it cannot encode the strict absence of edges between $C_1, C_2$ as the C-DAG does.

(ii) C-DAGs allow for bidirected edges between clusters, whereas TBK does not, because a bidirected edge could violate both conditions of Definition 2.

Looking at concrete examples, consider gene regulatory networks where the p53 pathway ($C_1$) and the Wnt pathway ($C_2$) both regulate cell cycle progression ($C_3$): $C_1 \to C_3 \leftarrow C_2$. TBK would require placing either $C_1$ or $C_2$ first in the tier ordering, which would incorrectly allow edges between p53 and Wnt pathway genes. In contrast, the C-DAG correctly encodes that $C_1$ and $C_2$ are non-adjacent while both cause $C_3$. Similar structures arise in climate science (multiple forcings affecting regional climate) and epidemiology (multiple risk factor domains affecting disease outcomes). We discuss application areas where C-DAG causal discovery is preferred over TBK based causal discovery in more detail in Appendix E.

## 3 Causal discovery with C-DAGs

In this section, we study how C-DAGs with and without unobserved confounding can be incorporated as prior knowledge to improve efficiency and accuracy of constraint-based causal discovery on the micro-variables. The core innovation of both Cluster-PC and Cluster-FCI, compared to PC/FCI is to prune the starting graph, and then, using the cluster structure, to reduce the set of possible separating variables. This reduces the number of CI tests the algorithm has to perform. The final edge-orientation phase is the same for all four algorithms.

### 3.1 Cluster-PC

First, we consider C-DAGs without latent confounding, neither within nor between clusters. First, we consider C-DAGs without latent confounding, neither within nor between clusters. The assumed C-DAG structure over the micro variables then allows for immediate pruning of the initial complete graph and also for partial orientation of edges via Algorithm 1. The resulting graph is an MPDAG (see Definition 11), which can contain both directed and undirected edges. We explain this procedure in more detail in Section A.1.

---
**Algorithm 1** C-DAG to MPDAG
---
**Require:** C-DAG $G_C = (V_C, E_C)$, $\mathcal{C} = \{C_1, \ldots, C_r\}$
 1: Form fully connected graph $G$ over $\cup_{i \in [r]} C_i$
 2: **for** $C_i, C_j \in \mathcal{C}$ **do**
 3:     **for** $X \in C_i, Y \in C_j$ **do**
 4:         **if** $C_i \to C_j$ **then** orient $X \to Y$ in $G$.
 5:         **if** $C_i \leftarrow C_j$ **then** orient $X \leftarrow Y$ in $G$.
 6:         **if** $C_i \not\to C_j$ **then** delete edge $X - Y$ in $G$.
 7: **return** MPDAG $G$.
---

The resulting graph after Algorithm 1 is an MPDAG (see Definition 11), which can contain both directed and undirected edges. We describe this procedure in detail in Section A.1. Instead of the fully connected graph, this MPDAG will serve as a starting point for our constraint based causal discovery algorithm (akin to PC). The reduced number of adjacencies and partially directed edges reduce the number of required conditional independence (CI) tests during the Cluster-PC algorithm in three ways: (i) directly, as non-adjacent variables are not tested for (conditional) independence anymore, (ii) reducing the number of potential separating sets

to be considered, and (iii) by working along a topological ordering of the clusters, more CI tests can be avoided. As an example for the latter, consider the C-DAG $C_1 \to C_2$. For $X, Y \in C_1$ we only need to consider separation sets $S \subset C_1$, as any path going through $C_2$ contains a collider in $C_2$, which is blocked when $C_2 \cap S = \emptyset$. More generally, the only candidates for a separating set between $X \in C_1$ and $Y \in C_2$ for the C-DAG $C_1 \gets C_2$ are subsets of the potential parents of $X$ in $C_1$ due to Proposition 1. We now define the general notion of relevant potential separation sets to consider.

**Definition 3 (Non-child).** *In a PDAG $G$, the non-children of a node $X$ is the set of adjacent nodes $adj_X^G$ of $X$ that are not children of $X$, i.e., $nch_X^G := adj_X^G \setminus ch_X^G$.*

We can now state the full Cluster-PC (C-PC) algorithm in Algorithm 2 and show that it is sound and complete.

**Theorem 2 (C-PC is sound and complete).** *Let $G_C$ be a C-DAG compatible with a DAG $G$. Then Algorithm 2, is sound and complete (for a CI oracle) in that it returns the same MPDAG obtained from running PC on $G$ and orienting all possible additional edges induced by the prior knowledge in $G_C$.*

The proof can be found in Section A.3. When a CI oracle is available, C-PC and PC with post-processing according to the C-DAG return the same result, so C-PC seemingly adds no value. However, in practice CI testing is an inherently difficult problem (Shah & Peters, 2020; Lundborg et al., 2022) and a whole line of works investigates how the number of CI tests can be reduced to render constraint based causal discovery more effective (Xie & Geng, 2008; Zhang et al., 2024; Shiragur et al., 2024). In Section 4 we demonstrate that avoiding unnecessary CI tests by incorporating the prior knowledge in C-PC indeed leads to substantial performance improvements.

**Definition 4 (Compatibility of CDPAGs and MPDAGs with C-DAGs).** *Let $G = (V, E)$ be the MPDAG obtained from applying Algorithm 1 to C-DAG $G_C$. A CPDAG $G' = (V', E')$ is called compatible with C-DAG $G_C$ if and only if $V' = V$ and for any $X, Y \in V'$ the following holds:*

*(i) If $(X - Y) \in E' \Rightarrow (X - Y) \in E$.*

*(ii) If $(X \to Y) \in E' \Rightarrow (X \to Y) \in E$ (analogous for $(X \gets Y) \in E'$).*

*A graph $G'$ being compatible with $G_C$ means it can be generated from an algorithm doing edge deletions and orientations on $G$.*

**Comparison to k-PC.** A relevant point of comparison to C-PC is k-PC (Kocaoglu, 2023). Both restrict the conditional independence tests used during constraint-based discovery in order to improve robustness and efficiency. The key difference is that k-PC imposes an explicit global bound $|S| \le k$ on conditioning sets, whereas C-PC constrains conditioning sets implicitly through the C-DAG: for a given pair of variables, only variables compatible with the cluster-level parent structure and current non-child relations are considered as candidate separators. Thus, C-PC does not fix a universal conditioning-set size in advance, but uses the cluster structure to exclude many irrelevant conditioning variables before testing.

## 3.2 Cluster-FCI

Next, we turn to the partially observed setting, where there may exist latent confounders between observed variables, represented by bidirected edges. Here, we allow bidirected edges both between the actual variables as well as between clusters of variables in the C-DAG. In Section 3.1, going from the C-DAG to an initial pruned and partially oriented graph was straightforward. This is no longer the case with latent variables, as the cluster graph may now be an ADMG as well (see Definition 13). We thus refer to such cluster graphs as C-ADMGs. Operating causal discovery directly on ADMGs is a dead end in that one can not distinguish whether there is one or two edges between a pair of nodes. Intuitively, if there may be latent confounders, it is generally impossible to determine whether observed dependence is due to a direct effect or due to latent confounding. In addition, in an ADMG nodes can be non-adjacent while still not being m-separable (m-separation is the natural extension of d-separation to graphs with bidirected edges, see Definition 16) due to the existence of inducing paths (Definition 19).

---

**Algorithm 2** Cluster-PC algorithm

---

**Require:** joint distribution $P_X$ over $d$ variables Markov and faithful w.r.t. the ground truth graph, CI oracle, C-DAG $G_C = (V_C, E_C), C = \{C_1, \dots, C_r\}$ with clusters in topological ordering and $G_C$ compatible with the ground truth CPDAG, see Definition 4.

1: Get MPDAG $G = (V, E)$ from Algorithm 1.       ▷ *pa, ch, an, de, nb, sib and nch refer to this current G*
2: **for** $m \in [r]$ **do**
3:    $L_m \leftarrow C_m \cup \bigcup_{C_s \in pa_{C_m}^{G_C}} C_s$
4:    **for** $k = 0, \dots, |L_m| - 2$ **do**
5:      $del_E \leftarrow \emptyset$
6:      **for** all pairs $X_j \in C_m$ and $X_i \in pa_{X_j}$ **do**
7:        **for** all $S \subset nch_{X_j} \setminus \{X_i\}$ with $|S| = k$ **do**
8:          **if** $X_i \perp\!\!\!\perp X_j \mid S$ **then**
9:            $del_E \leftarrow (X_i \rightarrow X_j)$
10:            $S_{\{i,j\}} \leftarrow S$
11:      **for** all adjacent $X_i, X_j \in C_m$ **do**
12:        **for** all $S \subset nch_{X_j} \setminus \{X_i\}$ or $S \subset nch_i \setminus \{X_j\}$ with $|S| = k$ **do**
13:          **if** $X_i \perp\!\!\!\perp X_j \mid S$ **then**
14:            $del_E \leftarrow \{(X_i \rightarrow X_j), (X_j \leftarrow X_i)\}$
15:            $S_{\{i,j\}} \leftarrow S$
16:      $E \leftarrow E \setminus del_E$
17: **for** each triple $X_i, X_j, X_k \in V$ with $X_i - X_j - X_k$, $X_i - X_j \leftarrow X_k$ or $X_i \rightarrow X_j - X_k$ and $X_i \not\sim X_k$ **do**       ▷ *find v-structures*
18:    **if** $X_j \notin S_{\{i,k\}}$ **then**
19:      orient the edges as $X_i \rightarrow X_j \leftarrow X_k$
20: Successively apply Meek's edge orientation rules, see Fig. 6.
21: **return** MPDAG $G = (V, E)$

---

**Algorithm 3** C-ADMG to partial mixed graph transformation for C-FCI

---

**Require:** C-ADMG $G_C$ with $C = \{C_1, \dots, C_r\}$

1: initialize $G$ as a complete undirected graph over $V = \bigcup_{i=1}^{r} C_i$
2: **for** all clusters $C_i$ in $G_C$ **do**
3:    **for** all $X, Y \in C_i$ **do**
4:      add $X \circ\!\!-\!\!\circ Y$ to $G$
5: **for** all adjacent clusters $C_i, C_j$ in $G_C$ **do**
6:    **for** all $X \in C_i, Y \in C_j$ **do**
7:      **if** $C_i \rightarrow C_j$ and $C_i \not\leftrightarrow C_j$ **then** $E \leftarrow E \cup \{X \rightarrow Y\}$
8:      **if** $C_i \leftarrow C_j$ and $C_i \not\leftrightarrow C_j$ **then** $E \leftarrow E \cup \{X \leftarrow Y\}$
9:      **if** $C_i \rightarrow C_j$ and $C_i \leftrightarrow C_j$ **then** $E \leftarrow E \cup \{X \circ\!\!\rightarrow Y\}$
10:      **if** $C_i \leftarrow C_j$ and $C_i \leftrightarrow C_j$ **then** $E \leftarrow E \cup \{X \leftarrow\!\!\circ Y\}$
11:      **if** $C_i \not\rightarrow C_j$, $C_i \not\leftarrow C_j$ and $C_i \leftrightarrow C_j$ **then** $E \leftarrow E \cup \{X \leftrightarrow Y\}$
12: **for** all non-adjacent clusters $C_i, C_j$ connected by an inducing path in $G_C$ **do**
13:    **if** $C_i \in an_{C_j}^{G_C}$ **then**
14:      **for** all $X \in C_i, Y \in C_j$ **do** $E \leftarrow E \cup \{X \rightarrow Y\}$
15:    **if** $C_j \in an_{C_i}^{G_C}$ **then**
16:      **for** all $X \in C_i, Y \in C_j$ **do** $E \leftarrow E \cup \{X \leftarrow Y\}$
17:    **if** $C_i \notin an_{C_j}^{G_C}$ and $C_j \notin an_{C_i}^{G_C}$ **then**
18:      **for** all $X \in C_i, Y \in C_j$ **do** $E \leftarrow E \cup \{X \leftrightarrow Y\}$
19: **return** partial mixed graph $G_{pm} := G$

---

Instead of ADMGs, the literature has converged on ancestral graphs (see Definition 17) as the core objects in constraint based causal discovery with latent confounders. In our case, we first derive a partial mixed graph from the given C-ADMG and operate our Cluster-FCI (C-FCI) algorithm on it, which ultimately outputs a partial ancestral graph (see, Definition 21). This output can be viewed as the analogue of the CPDAG in the fully observed setting.

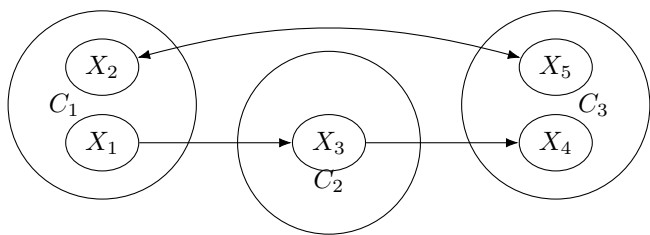

Figure 2: Example of non-ancestral C-ADMG.

**Definition 5** (**Partial mixed graph for C-ADMG**)**.** *The partial mixed graph $G_{pm} = (V, E)$ of a C-ADMG $G_C$ is a graph (with four types of possible edges $\rightarrow$, $\leftrightarrow$, $\circ\!-\!\circ$, $\circ\!\rightarrow$), such that for all clusters $C_i, C_j$*

*(i) for all $X, Y \in C_i$ we have $(X \circ\!-\!\circ Y) \in E$,*

*(ii) for all $X \in C_i, Y \in C_j$ with $C_i \rightarrow C_j$ and $C_i \not\leftrightarrow C_j$ we have $(X \rightarrow Y) \in E$,*

*(iii) for all $X \in C_i, Y \in C_j$ with $C_i \rightarrow C_j$ and $C_i \leftrightarrow C_j$ we have $(X \circ\!\rightarrow Y) \in E$,*

*(iv) for $X \in C_i, Y \in C_j$ with $C_i, C_j$ not adjacent and connected by an inducing path, if*

- *$C_i \in an_{C_j}^{G_C}$ we have $(X \rightarrow Y) \in E$,*
- *$C_j \in an_{C_i}^{G_C}$ it is $(X \leftarrow Y) \in E$,*
- *$C_i \notin an_{C_j}^{G_C}$ and $C_j \notin an_{C_i}^{G_C}$ we have $(X \leftrightarrow Y) \in E$.*

*The circle indicates uncertainty on the nature of the edge mark; it could be a tail or an arrow. In MPDAGs, an undirected edge $X - Y$ was used to express uncertainty about the edge direction. In partial mixed graphs this function is now served by the edge $X \circ\!-\!\circ Y$.*

For brevity of notation, as in Zhang (2008b), we also introduce the "$*$" symbol to denote either an arrowhead, circle or tail. This will be used later for edge orientations, where some edge marks don't matter for the orientation rule. For example if the edge $X *\!-\!\circ Y$ is oriented as $X *\!\rightarrow Y$, the edge mark on the left stays whatever it was before the orientation. In the original definition of partial ancestral graphs (Zhang, 2008b), undirected edges "$-$" or "$\circ\!-$" are present in order to consider potential selection variables. In our work we assume non-existence of selection variables, so we omit these undirected edges here.

The partial mixed graph $G_{pm}$ for a C-ADMG $G_C$ will be the starting graph for the Cluster-FCI Algorithm 4 and is produced from $G_C$ by using Algorithm 3. To obtain an ancestral graph later on during C-FCI, non-adjacent but non-m-separable nodes need to have an edge introduced between them in the preprocessing, see Algorithm 3(l. 12-18).

**Definition 6** (**Compatibility of MAGs and partial mixed graphs with C-ADMGs**)**.** *Let $G_{pm} = (V, E)$ be the partial mixed graph obtained from applying Algorithm 3 to C-ADMG $G_C$. A partial mixed graph $G'_{pm} = (V', E')$ is called compatible with C-ADMG $G_C$ if $V' = V$ and for any $X, Y \in V'$ the following holds:*

*(i) $(X \circ\!-\!\circ Y) \in E' \Rightarrow (X \circ\!-\!\circ Y) \in E$,*

*(ii) $(X \circ\!\rightarrow Y) \in E' \Rightarrow \{(X \circ\!-\!\circ Y), (X \circ\!\rightarrow Y)\} \cap E \neq \emptyset$ (analogous for $(X \leftarrow\!\circ Y) \in E'$),*

*(iii) $(X \leftrightarrow Y) \in E' \Rightarrow \{(X \circ\!-\!\circ Y), (X \circ\!\rightarrow Y), (X \leftarrow\!\circ Y), (X \leftrightarrow Y)\} \cap E \neq \emptyset$,*

*(iv) $(X \rightarrow Y) \in E' \Rightarrow X \in nch_Y^{G_{pm}}$ (analogous for $(X \leftarrow Y) \in E'$).*

*Compatibility of MAGs (which are partial mixed graphs that satisfy ancestrality and maximality (Zhang, 2008b)) with C-ADMGs follows directly from this definition, too. To summarize, a graph $G'_{pm}$ being compatible with $G_C$ means it can be generated from an algorithm doing edge deletions and orientations on $G_{pm}$.*

The graph resulting from Algorithm 3 need not necessarily be ancestral yet, as the input C-ADMG, and thus also the output, may contain almost directed cycles. An almost directed cycle is defined as $X \leftrightarrow Y$ and

---

**Algorithm 4** Cluster-FCI algorithm

---

**Require:** Joint distribution $P_O$ of $d$ observed variables, independence oracle, C-DAG $G_C = (V_C, E_C), C = \{C_1, \ldots, C_r\}$ with clusters in topological ordering (w.r.t. directed edges) compatible with ground truth MAG, see Definition 6.

1: Construct graph $G = (V, E)$ from $G_C$ with Algorithm 3.
2: **for** $m \in [r]$ **do**
3:    $L_m \leftarrow C_m \cup \bigcup_{C_s \in pa_{C_m}^{G_C}} C_s \cup \bigcup_{C_s \in sib_{C_m}^{G_C}} C_s$
4:    **for** $k = 0, \ldots, |L_m| - 2$ **do**
5:       $del_e \leftarrow \emptyset$
6:       **for** for all $X_i \in C_m$ and $X_j \in nch_{X_i}$ **do**
7:          **for** all $S \subset nch_{X_j} \setminus \{X_i\}$ or $S \subset nch_{X_i} \setminus \{X_j\}$ with $|S| = k$ **do**
8:             **if** $X_i \perp\!\!\!\perp X_j \mid S$ **then**
9:                $del_E \leftarrow del_E \cup \{X_i *\!\!-\!\!* X_j\}$
10:                $S_{\{i,j\}} \leftarrow S$
11:       $V \leftarrow V \setminus del_E$
12: **for** all unshielded triples $(X_i, X_j, X_k)$ **do**
13:    **if** $X_j \notin S_{\{i,k\}}$ **then**
14:       **if** $X_i *\!\!-\!\!\circ X_j \circ\!\!-\!\!* X_k$ **then** orient $X_i *\!\!-\!\!\circ X_j \circ\!\!-\!\!* X_k$ as $X_i *\!\!\rightarrow X_j \leftarrow\!\!* X_k$ in $G$
15:       **if** $X_i *\!\!\rightarrow X_j \circ\!\!\rightarrow X_k$ **then** orient $X_i *\!\!\rightarrow X_j \circ\!\!\rightarrow X_k$ as $X_i *\!\!\rightarrow X_j \leftrightarrow X_k$ in $G$
16: **for** all $X_i \in V$ **do**
17:    **for** all $X_j \in adj_{X_i}^G$ **do**
18:       compute $pds(X_i, X_j)$ as in Definition 22.
19:       **for** $k = 0, \ldots, d - 2$ **do**
20:          **for** $|S| \subset pds(X_i, X_j)$ with $|S| = k$ **do**
21:             **if** $X_i \perp\!\!\!\perp X_j \mid S$ **then**
22:                $V \leftarrow V \setminus \{X_i *\!\!-\!\!* X_j\}$
23:                $S_{\{i,j\}} \leftarrow S$
24: Reorient all edges according to C-DAG $G_C$ (as in Algorithm 3 but only orienting edges, not adding edges)
25: For any almost directed cycle $X_l \leftrightarrow X_i \rightarrow \ldots \rightarrow X_l$ orient $X_l \leftrightarrow X_i$ to $X_l \rightarrow X_i$
26: Use rules R0-R4, R8-R10 of (Zhang, 2008b) to orient as many edge marks as possible.
27: **return** PAG $G = (V, E)$

---

$X \in an_Y^G$ (Zhang, 2008b). A direct conversion of the C-ADMG to a MAG, e.g., following Hu & Evans (2020), is undesirable, as the following example demonstrates. Consider the graph in Fig. 2. The MAG $G_M$ over $\{X_1, \ldots, X_5\}$ is ancestral, but the corresponding C-ADMG $G_C$ with $C = \{C_1, C_2, C_3\}, C_1 = \{X_1, X_2\}, C_2 = \{X_3\}, C_3 = \{X_4, X_5\}$ is not, due to the almost directed cycle $C_3 \leftrightarrow C_1 \rightarrow C_2 \rightarrow C_3$. Even though there is an almost directed cycle in $G_C$, $G_M$ is compatible with $G_C$. If we were to transform this C-ADMG into a MAG, the edge $C_1 \leftrightarrow C_2$ would change to $C_1 \rightarrow C_2$. The edge between $X_2, X_5$ would consequently inherit this orientation as $X_2 \rightarrow X_5$ contradicting the correct edge type $X_2 \leftrightarrow X_5$. Therefore, we only reorient almost directed cycles at the very end of C-FCI, ensuring it outputs a valid PAG without introducing false edge information.

**Definition 7** (**Updated pa, ch and nch, and sib**)**.** *In addition to the previous definitions, in the following, whenever $X \circ\!\!\rightarrow Y$, we count $X$ as a parent of $Y$, $X \in pa_Y$, and vice versa $Y \in ch_X$. Whenever $X \circ\!\!-\!\!\circ Y$, $X \circ\!\!\rightarrow Y$, or $X \leftrightarrow Y$, we count $X$ as a non-child of $Y$, $X \in nch_Y$ and these definitions extend to ancestors and descendants. Whenever $X \leftrightarrow Y$, we call $X$ is a sibling of $Y$, $X \in sib_Y$ (and vice versa).*

Cluster-FCI follows a similar strategy as Cluster-PC by running CI tests on a per-cluster basis (while following the general logic of FCI) and is developed in all detail in Algorithm 4.

We now highlight some differences between Cluster-FCI and FCITiers. Cluster-FCI starts with the first cluster of the topological ordering and works its way down along the topological ordering, while FCITiers (Andrews et al., 2020, Alg. 1) starts with the last and works its way up. However, due to the nature of FCITiers, this direction does not matter as it is running a version of FCI on disjoint sets of edges (FCIExogenous). These disjoint edge sets are derived from the TBK. The resulting edges from FCIExogenous are then added to the overall graph. On the contrary, Cluster-FCI pre-processes the entire fully connected

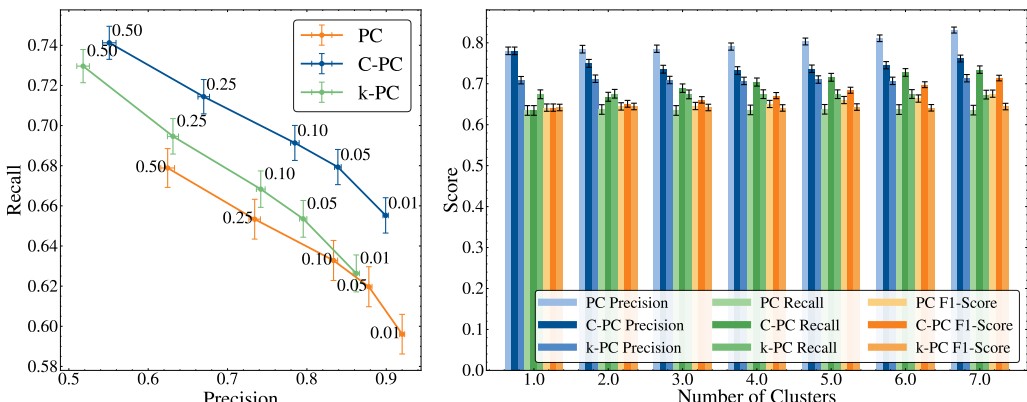

Figure 3: Comparison of PC, k-PC ($k = 2$) and C-PC, adjacency metrics with $\pm 95\%$ CI interval based on t-distribution. **Left:** Precision vs. recall for different significance levels $\alpha$ of the CI test. Cluster-PC dominates base PC w.r.t. recall and for common values of $\alpha \in \{0.05, 0.01\}$ achieves substantially better recall with only small reductions in precision. Cluster-PC also dominates k-PC for both precision and recall. **Right:** Precision, recall and F1-score for different numbers of clusters. The F1-score of C-PC increases with the number of clusters, which amounts to more granular background knowledge.

graph according to the C-ADMG to obtain a partial mixed graph. It then removes further edges along the topological ordering. Our proposed C-FCI Algorithm 4 is sound, but not complete (proof in Section A.4).

**Theorem 3** (**Soundness of Cluster-FCI**). *If the C-DAG $G_C$ is compatible with ground truth MAG $G_M$, C-FCI is sound in the sense that nodes $X_i, X_j$ are adjacent in the output PAG $G_P$ if and only if they are adjacent in the ground truth MAG $G_M$. In addition, all arrow and tail edge marks in $G_P$ are also present in $G_M$.*

**Too much causal information—incompleteness of C-FCI.** The example in Fig. 2 shows that a C-ADMG can encode arrowheads that contradict ancestrality (imagine an additional edge $X_1 \leftrightarrow X_4$). C-FCI can be adjusted to not output a PAG (Non-PAG C-FCI), by not re-orienting almost directed cycles. This variation can be an improvement, as it increases the information contained in the obtained graph. To see this, consider the graph in Fig. 2 with an additional edge $X_1 \leftrightarrow X_4$. C-FCI would return the edge $X_1 \rightarrow X_4$, due to the almost directed cycle $X_4 \leftrightarrow X_1 \rightarrow X_3 \rightarrow X_4$. Non-PAG C-FCI in contrast will return $X_1 \leftrightarrow X_4$, the correct and more informative result. Further exploring this deviation from the well-explored setting of relying primarily on ancestral graphs for causal discovery is an interesting direction for future work.

This example simultaneously highlights that C-FCI is incomplete: its output does not always reveal all determined causal information. In a way, C-FCI is "overwhelmed by prior causal information," as some (useful) background knowledge can violate ancestrality and thus not be captured properly in the algorithm. However, C-FCI always remains at least as informative as FCI. In Remark 1 we additionally sketch that C-FCI is also at least as informative as FCITiers for a suitable set of tiers—recalling that C-DAGs are strictly more flexible then TBK. Finally, we conjecture that for ancestral C-ADMGs, C-FCI is also complete, i.e., all counterexamples have to rely on non-ancestral C-ADMGs as background knowledge. This also remains an open question for future work.

## 4 Simulation studies

We now empirically demonstrate the differences between PC vs. Cluster-PC vs. k-PC ($k = 2$) (Kocaoglu, 2023)and FCI vs. FCITiers vs. Cluster-FCI in different simulation studies. All code for these experiments is available on Github[1]. In the first setting, we sample 12600 Erdős–Rényi graphs and vary the number of nodes, edges, and the significance level $\alpha$ for the chosen CI tests. We compare Cluster-PC to PC and k-PC

---

[1] https://github.com/JanMarcoRuizdeVargas/clustercausal/

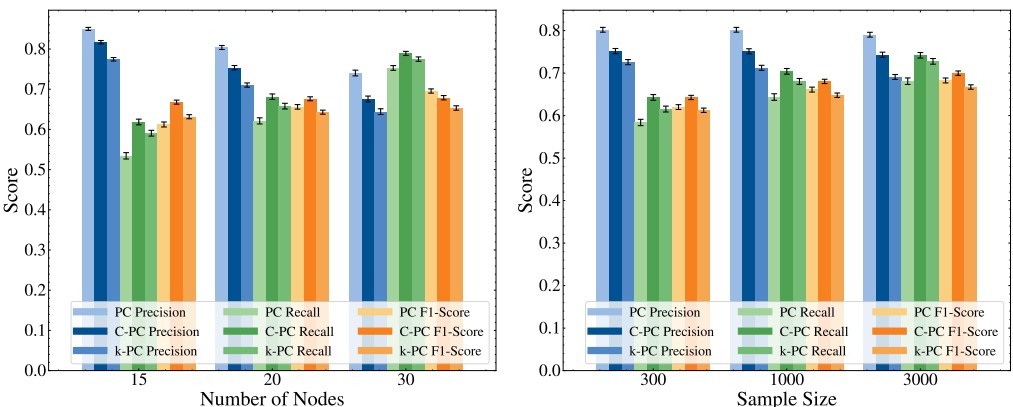

Figure 4: Comparison of PC, k-PC ($k = 2$) and C-PC w.r.t. adjacency precision, and F1-score. Metrics with ±95% CI interval based on t-distribution. **Left:** Graphs with different number of nodes. Smaller graphs are denser (same number of edges) and the algorithms have better precision, compared to on larger graphs having better recall. Cluster-PC trades off precision to recall compared to PC and is strictly better than k-PC. **Right:** Differing sample size, larger sample sizes have a small positive impact on recall.

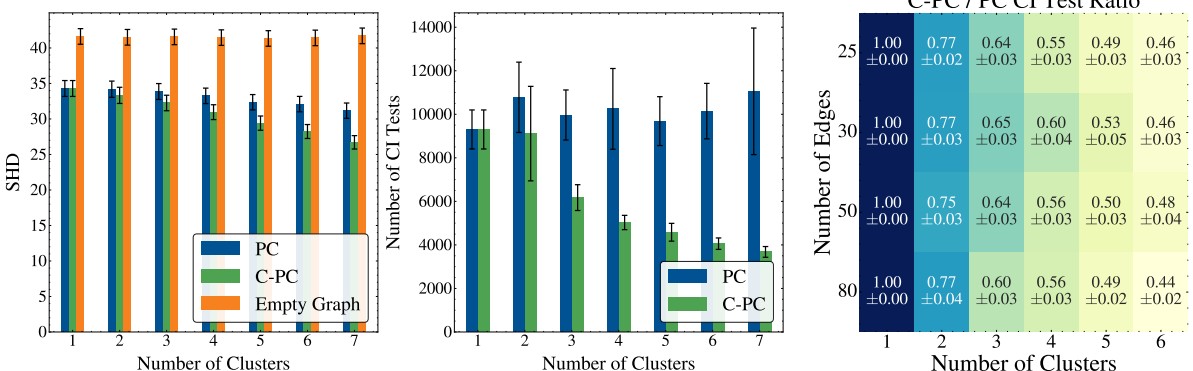

Figure 5: Metrics with ±95% CI interval based on t-distribution. **Left:** Structural Hamming distance (SHD) for different numbers of clusters. The empty graph is used as a dummy reference. Again, C-PC clearly benefits from a growing number of clusters. **Middle:** The number of conditional independence (CI) tests is substantially reduced in C-PC for C-DAGs with many clusters. Notably, even for coarse background knowledge of only two clusters, the number of required CI tests already drops noticeably. **Right:** The ratio of CI tests between PC and C-PC (for the same graph, across different numbers of edges and clusters), highlights that the savings remain roughly constant for a fixed number of clusters even as the number of edges increases.

here. The second simulation study performs sensitivity analysis w.r.t. different graph generation methods and probability distributions. The third compares the three FCI variants using generated C-ADMGs (that not necessarily satisfy TBK). The last simulation study compares the FCI variants on C-DAGs, which do satisfy TBK. Section B.2 contains a detailed breakdown of all chosen simulation parameters and settings.

We evaluate the discovery algorithms with respect to precision, recall, F1-score, Structural Hamming Distance (SHD), and the number of required CI-tests. For these metrics, we distinguish between 'adjacency', i.e., is there any edge present between two variables, and 'arrow', which compares the types of edge marks. Detailed definitions of the used metrics can be found in Appendix B. We also run the algorithms with different significance levels $\alpha$. Since rejecting the null hypothesis, i.e., rejecting conditional independence, leads to an edge *not* being removed, while failure to reject leads to a removed edge, higher $\alpha$ leads to fewer edge deletions and overall denser graphs.

| Metric | PC | Cluster-PC | k-PC |
|---|---|---|---|
| Adj. precision | **79.8% ± 0.3%** | 74.9% ± 0.3% | 71.0% ± 0.3% |
| Arrow precision | 54.6% ± 0.4% | **66.6% ± 0.4%** | 40.5% ± 0.3% |
| Adj. recall | 63.6% ± 0.4% | **69.6% ± 0.4%** | 67.4% ± 0.4% |
| Arrow recall | 33.9% ± 0.3% | **58.7% ± 0.4%** | 55.2% ± 0.4% |
| Adj. F1-score | 65.5% ± 0.3% | **67.4% ± 0.3%** | 64.3% ± 0.3% |
| Arrow F1-score | 38.5% ± 0.3% | **58.2% ± 0.4%** | 42.6% ± 0.2% |
| SHD | 33.0 ± 0.4 | **30.7 ± 0.4** | 46.9 ± 0.5 |
| Avg. CI tests | 10170 ± 630.5 | 5991 ± 359.4 | **2567 ± 50.3** |

Table 1: Comparison of metrics for Simulations 1 (±95%CI-interval based on t-distribution). C-PC considerably outperforms PC in the arrow metrics. The cluster version is slightly worse in adjacency precision, but shows improvements in adjacency recall, F1-score, SHD and reduces the number of CI tests required. C-PC is better than k-PC in all metrics, while requiring double the number of CI tests.

| Metric | FCI | C-FCI | FCITiers |
|---|---|---|---|
| Adj. precision | **29.2% ± 1.1%** | 28.2% ± 1.1% | 26.6% ± 1.1% |
| Arrow precision | 20.4% ± 0.9% | 24.1% ± 0.9% | **25.2% ± 1.0%** |
| Adj. recall | 17.6% ± 0.7% | **19.7% ± 0.8%** | 18.4% ± 0.7% |
| Arrow recall | 10.1% ± 0.5% | **16.1% ± 0.7%** | 13.7% ± 0.6% |
| Adj. F1-score | 21.2% ± 0.8% | **22.5% ± 0.8%** | 21.0% ± 0.8% |
| Arrow F1-score | 12.8% ± 0.6% | **18.6% ± 0.7%** | 17.0% ± 0.7% |
| SHD | **38.4 ± 0.7** | 40.5 ± 0.7 | 39.8 ± 0.7 |
| Avg. CI tests | 1585 ± 46.6 | **918 ± 31.4** | 1746 ± 46.3 |

Table 2: Comparing FCI, Cluster-FCI, and FCITiers for Simulations 3, which generated C-ADMGs, which do not necessarily satisfy TBK. The clusters were generated using the "dag-first" generation method, see Section B.3. The methods are overall very close in performance, but C-FCI needs significantly less CI tests.

Figs. 3 and 4 and Table 1 demonstrate that C-PC dominates PC in recall as well as arrow precision and F1-score. The disadvantage in adjacency precision is made up by gains in recall and results in a higher F1-score for C-PC over PC. Fig. 5 further shows that the efficiency, i.e., the number of CI tests, drastically improved with more fine grained background knowledge (more clusters), while also improving the overall performance of the causal discovery measured by SHD. Even coarse background knowledge from a C-DAG consisting of just two clusters substantially reduces the number of required CI tests. Finally, the efficiency gains are not sensitive to the overall number of edges in a graph, but only depend on the number of clusters, i.e., the 'amount of the background information.' Compared to k-PC ($k = 2$), C-PC has consistently better metrics, while not reducing CI tests as drastically as k-PC, see Table 1.

| Metric | FCI | C-FCI | FCITiers |
|---|---|---|---|
| Adj. precision | **92.1% ± 0.4%** | 89.9% ± 0.5% | 89.7% ± 0.5% |
| Arrow precision | 70.5% ± 0.7% | **81.3% ± 0.6%** | 80.5% ± 0.6% |
| Adj. recall | 71.0% ± 0.7% | **71.7% ± 0.7%** | **71.7% ± 0.7%** |
| Arrow recall | 46.4% ± 0.8% | 50.1% ± 0.7% | **50.9% ± 0.7%** |
| Adj. F1-score | **79.1% ± 0.5%** | 78.6% ± 0.4% | 78.4% ± 0.4% |
| Arrow F1-score | 54.5% ± 0.6% | 60.8% ± 0.6% | **61.1% ± 0.6%** |
| SHD | 19.4 ± 0.3 | **16.2 ± 0.3** | 16.8 ± 0.3 |
| Avg. CI tests | 1686 ± 47.0 | **959 ± 30.6** | 1775 ± 45.5 |

Table 3: Comparing FCI, Cluster-FCI, and FCITiers for Simulation 4. Simulation 4 generated C-DAGs with latent variables only within clusters, using the "cdag-first" method, see Section B.3. The methods are overall very close in performance, but C-FCI needs significantly less CI tests.

For C-FCI, Table 2 shows improved accuracy and arrow precision with minor hits in adjacency precision compared to FCI and FCITiers in the setting where C-DAGs do not necessarily satisfy TBK. While overall SHD remains comparable, C-FCI requires around half the number of CI tests. When generated C-DAGs only have latent variables within clusters, i.e., TBK can be satisfied according to some topologically ordered completion of the C-DAG (Table 3), C-FCI and FCITiers perform similarly (both generally outperforming vanilla FCI) whereas C-FCI again requires only about half the number of CI tests.

## 5  Conclusion

We leverage C-DAGs as a flexible and realistic type of background knowledge for constraint-based causal discovery in fully and partially observed settings. C-DAGs are a provably superior alternative for encoding group-wise background knowledge compared to the existing tiered background knowledge. We develop the Cluster-PC (C-PC) and Cluster-FCI (C-FCI) algorithms and prove that they are complete and sound or sound (but not complete) respectively. The non-completeness of C-FCI is shown to stem from C-ADMGs possibly containing 'more causal information' than the well-established PAG representation in causal discovery with latents can handle. Through extensive empirical simulation studies we demonstrate that our proposed algorithms indeed outperform the corresponding algorithms with no, or existing types of background knowledge across a wide range of settings on most metrics.

Interesting future directions for future work include to apply C-DAG background knowledge to score-based methods (see Appendix D for some first thoughts), to combine C-DAGs with other types of prior knowledge such as pairwise background knowledge, or to include selection variables in the causal discover process as well. Since C-ADMGs can contain background knowledge that violates ancestrality required for FCI, it will also be interesting to investigate under which types of background knowledge an extended FCI version remains complete. Lastly, using summary causal graphs (Ferreira & Assaad, 2025b)—also allowing for cycles—instead of C-DAGs for improved causal discovery is also an interesting direction for follow up work.

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

# A  Definitions, theorems, additional explanations

## A.1  A C-DAG leads to an MPDAG

The Markov equivalence class of a graph $G$, the graphs entailing the same d-separations, can be characterized by a CPDAG.

**Definition 8** (**CPDAG (completed partially directed graph)**, Andersson et al., 1997)**.** *The completed partially directed graph of a DAG $G$, denoted by $G^*$, is a graph with the same skeleton as $G$ and undirected edges. A directed edge occurs if and only if that directed edge is present in all DAGs of the Markov equivalence class of $G$. Directed edges come from either v-structures or applying the orientation rules in Fig. 6.*

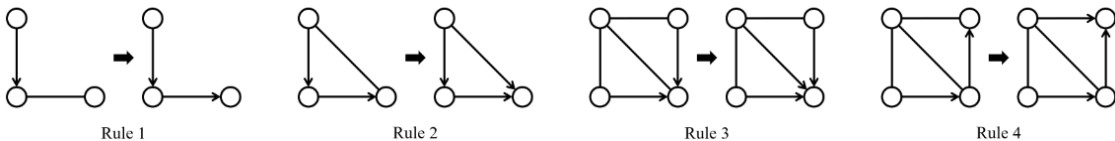

Rule 1  Rule 2  Rule 3  Rule 4

Figure 6: Meek's orientation rules (Meek, 1995) (Figure from (Fang et al., 2025)).

**Definition 9** (**Pairwise Causal Constraints**, Fang et al., 2025)**.** *A direct causal constraint, denoted by $X \rightarrow Y$, is a proposition stating that $X$ is a parent of $Y$, i.e., $X$ is a direct cause of $Y$. An ancestral causal constraint, denoted by $X \dashrightarrow Y$, is a proposition stating that $X$ is an ancestor of $Y$, i.e., $X$ is a cause of $Y$. A non-ancestral causal constraint, denoted by $X \nrightarrow Y$, is a proposition stating that $X$ is not an ancestor of $Y$, i.e., $X$ is not a cause of $Y$. In all these cases, $X$ is called the* tail *and $Y$ is called the* head*.*

**Definition 10** (**Restricted Markov equivalence class**, Fang et al., 2025)**.** *The restricted Markov equivalence class induced by a CPDAG $G^*$ and a pairwise causal constraint set $B$ over $V$, denoted by $[G^*, B]$ is composed of all equivalent DAGs in $M(G^*)$ that satisfy $B$ ($M(G^*)$ is the Markov equivalence class of $G^*$).*

**Definition 11** (**Maximally partially directed acyclic graph (MPDAG)**, Fang et al., 2025)**.** *The MPDAG $H$ of a non-empty restricted Markov equivalence class $[G^*, B]$, induced by a CPDAG $G^*$ and a pairwise causal constraint set $B$ is a PDAG such that*

*(i) $H$ has the same skeleton and v-structures as $G^*$ and*

*(ii) an edge is directed in $H$ if and only if it appears in all DAGs in $[G^*, B]$.*

So for a given CPDAG, the C-DAG pairwise constraint set would be (also see Appendix C):

**Definition 12** (**Pairwise causal constraint set from C-DAGs**)**.** *The pairwise causal constraint $B$ set induced by C-DAG $G_C$ is as follows: For $X_i \in C_i, X_j, \in C_j, C_i \neq C_j$,*

- *if $C_i \rightarrow C_j$, then $X_i \nleftarrow X_j \in B$*
- *if $C_i \dashrightarrow C_j, C_i \nrightarrow C_j$, then $X_i \nleftarrow X_j, X_i \nleftarrow X_j \in B$*
- *if $C_i \nleftarrow C_j, C_i \nrightarrow C_j$, then $X_i \nleftarrow X_j, X_i \dashrightarrow X_j \in B$*

**C-DAG leads to MPDAG**. It follows from the definitions that a CPDAG restricted by a compatible C-DAG leads to an MPDAG. In the same way, restricting a fully connected graph with a C-DAG via Algorithm 1 also leads to an MPDAG, whose edges are a super-set of any compatible DAG or CPDAG (interpreting undirected edges as $\rightarrow$ *and* $\leftarrow$).

## A.2  Further definitions, theorems and additional explanations

**Definition 13** (**ADMG**)**.** *A directed mixed graph $G = (V, E)$ consists of a finite set of nodes $V$ and a finite set of edges $E$, which are either directed ($\rightarrow$) or bidirected ($\leftrightarrow$). An acyclic directed mixed graph (ADMG) is a directed mixed graph without directed cycles.*

**Definition 14** (**d-separation in C-DAGs**,Anand et al., 2023)**.** *A path $p$ in a C-DAG $G_C$ is said to be d-separated by a set of clusters $Z \subset C$ if and only if $p$ contains a triplet*

(i) $C_i ** C_m \to C_j$ *such that the non-collider cluster $C_m$ is in $Z$, or*

(ii) $C_i \!*\!\!\to C_m \leftarrow\!* C_j$ *such that the collider cluster $C_m$ and its descendants are not in $Z$.*

*A set of clusters $Z$ is said to d-separate two sets of clusters $X, Y \subset C$, denoted by $X \perp\!\!\!\perp_{G_C} Y \mid Z$, if and only if $Z$ blocks every path from a cluster in $X$ to a cluster in $Y$.*

**Definition 15** (**mns (minimal neighbor separator)**, Gupta et al., 2023)**.** *For a DAG $G = (V, E)$ and node $X$ and $A \notin nb_X^+$ ($nb_X^+ := nb_X \cup \{X\}$), the minimal neighbor separator $mns_X(A) \subset nb_X$ is the unique set of nodes such that*

(i) *(d-separation)* $A \perp\!\!\!\perp_G X \mid mns_X(A)$

(ii) *(minimality) for any $S \subset mns_X(A) : A \not\perp\!\!\!\perp_G X \mid S$*

*hold.*

**Proposition 1** (**Restricting separating set via mns**,Gupta et al., 2023)**.** *For any node $Y \notin de_X \cup nb_X^+$, the minimum neighbor separator $mns_X(Y)$ exists and $mns_X(Y) \subset pa_X$.*

**Definition 16** (**M-connecting, m-separation**)**.** *Let $G = (V, E)$ be a directed mixed graph (a graph containing directed and bidirected edges). A path between $X_i, X_j \in V$ is called m-connecting in $G$ given $S \subset V$ if every non-collider on the path is not in $S$, and every collider on the path is in $S$ or is an ancestor of $S$ in $G$. If there is no path m-connecting $X_i$ and $X_j$ in $G$ given $S$, $X_i$ and $X_j$ are called m-separated given $S$. Sets $A$ and $B$ are said to be m-separated given $S$, if for all $X_i \in A$ and all $j \in B$, $X_i$ and $X_j$ are m-separated given $S$.*

**Definition 17** (**Ancestral graph**)**.** *A mixed graph $G$ (containing directed and bidirected edges) is ancestral if the following three conditions hold:*

(i) *there is no directed cycle,*

(ii) *there is no almost directed cycle,*

(iii) *for any undirected edge $X_1 - X_2$, $X_1$ and $X_2$ have no parents or siblings.*

**Definition 18** (**MAG (maximal ancestral graph)**)**.** *An ancestral graph is called maximal if for any two non-adjacent vertices, there is a set of vertices that m-separates them.*

**Definition 19** (**Inducing path**,Zhang, 2008b)**.** *In an ancestral graph, let $X, Y$ be any two vertices and $L, S$ be disjoint sets of vertices not containing $X, Y$. $L$ describes the latent variables and $S$ describes the selection variables. A path $\pi$ between $X$ and $Y$ is called an inducing path relative to $(L, S)$ if every non-endpoint vertex on $\pi$ is either in $L$ or a collider, and every collider on $\pi$ is an ancestor of either $X, Y$, or a member of $S$. When $L = S = \emptyset$, $\pi$ is called a primitive inducing path between $X$ and $Y$.*

Two DAGs are Markov equivalent if and only if they have the same adjacencies and unshielded colliders. For two MAGs, this is a necessary condition, but not sufficient anymore. For two MAGs to be Markov equivalent, they also need to possess the same colliders on discriminating paths.

**Definition 20** (**Discriminating path**,Zhang, 2008b)**.** *In a MAG, a path between $X$ and $Y$, $\pi = (X, \dots, W, S, Y)$ is a discriminating path for $S$ if*

(i) $\pi$ *includes at least three edges,*

(ii) $S$ *is a non-endpoint vertex on $\pi$ and is adjacent to $Y$ on $\pi$,*

(iii) $X$ *is not adjacent to $Y$ and every vertex between $X$ and $S$ is a collider on $\pi$ and is a parent of $Y$.*

**Proposition 2** (**Markov equivalence criterion for MAGs**,Zhang, 2008b)**.** *Two MAGs over the same set of vertices are Markov equivalent if and only if*

*(i) they have the same adjacencies,*

*(ii) they have the same unshielded colliders,*

*(iii) if a path $\pi$ is a discriminating path for a vertex $S$ in both graphs, then $S$ is a collider on the path in one graph if and only if it is a collider on the path in the other.*

The partial ancestral graph is the CPDAG analogue for characterizing the Markov equivalence class:

**Definition 21** (**Partial ancestral graph**, Zhang, 2008b)**.** *Let $M(G)$ be the Markov equivalence class of a MAG $G$. A partial ancestral graph (PAG) for $M(G)$ is a graph $G_P$ with possibly three kind of edge marks (and hence six kinds of edges: $-$, $\rightarrow$, $\leftrightarrow$, $\circ\!-$, $\circ\!-\!\circ$, $\circ\!\rightarrow$), such that*

*(i) $G_P$ has the same adjacencies as $G$ (and any member of $M(G)$ and*

*(ii) every non-circle mark in $G_P$ is an invariant mark in $M(G)$.*

*If furthermore every circle in $G_P$ corresponds to a variant mark in $M(G)$, $G_P$ is called the maximally informative PAG for $M(G)$.*

In ancestral graphs, it may be possible that two m-separable nodes can not be m-separated by (a subset of) their neighbors. So FCI needs to search "possible d-separating sets" too, i.e., sets that contain nodes not adjacent to $X, Y$, but whose nodes may be necessary to m-separate $X$ and $Y$.

**Definition 22** (**Possible d-separating set**, Andrews et al., 2020)**.** *$X \in pds(X_i, X_j)$ if and only if $X \notin \{X_i, X_j\}$ and there is a path $\pi$ between $X_i$ and $X$ in $G$ such that for every subpath $(X_k, X_l, X_m)$ of $\pi$ either $X_l$ is a collider on $\pi$ or $X_k$ and $X_m$ are adjacent.*

**Definition 23** (**Orientation rules for FCI**, Zhang, 2008b)**.**

- **R0:** *For each unshielded triple $\langle \alpha, \gamma, \beta \rangle$ in $P$, orient as a collider $\alpha *\!\!\rightarrow \gamma \leftarrow\!\!* \beta$ if and only if $\gamma \notin Sepset(\alpha, \beta)$.*

- **R1:** *If $\alpha *\!\!\rightarrow \beta \circ\!\!-\!\!* \gamma$, and $\alpha$ and $\gamma$ are not adjacent, then orient the triple as $\alpha *\!\!\rightarrow \beta \rightarrow \gamma$.*

- **R2:** *If $\alpha \rightarrow \beta *\!\!\rightarrow \gamma$ or $\alpha *\!\!\rightarrow \beta \rightarrow \gamma$, and $\alpha \circ\!\!-\!\!\circ \gamma$, then orient $\alpha *\!\!-\!\!\circ \gamma$ as $\alpha *\!\!\rightarrow \gamma$.*

- **R3:** *If $\alpha *\!\!\rightarrow \beta \leftarrow\!\!* \gamma$, $\alpha *\!\!-\!\!\circ \theta \circ\!\!-\!\!* \gamma$, $\alpha$ and $\gamma$ are not adjacent, and $\theta *\!\!-\!\!\circ \beta$, then orient $\theta *\!\!-\!\!\circ \beta$ as $\theta *\!\!\rightarrow \beta$.*

- **R4:** *If $u = \langle \theta, \ldots, \alpha, \beta, \gamma \rangle$ is a discriminating path between $\theta$ and $\gamma$ for $\beta$, and $\beta \circ\!\!-\!\!* \gamma$; then if $\beta \in Sepset(\theta, \gamma)$, orient $\beta \circ\!\!-\!\!* \gamma$ as $\beta \rightarrow \gamma$; otherwise orient the triple $\langle \alpha, \beta, \gamma \rangle$ as $\alpha \leftrightarrow \beta \leftrightarrow \gamma$.*

- **R8:** *If $\alpha \rightarrow \beta \rightarrow \gamma$ or $\alpha -\!\!\circ \beta \rightarrow \gamma$, and $\alpha \circ\!\!\rightarrow \gamma$, orient $\alpha \circ\!\!\rightarrow \gamma$ as $\alpha \rightarrow \gamma$.*

- **R9:** *If $\alpha \circ\!\!\rightarrow \gamma$, and $p = \langle \alpha, \beta, \theta, \ldots, \gamma \rangle$ is an uncovered p.d. (partially directed) path from $\alpha$ to $\gamma$ such that $\gamma$ and $\beta$ are not adjacent, then orient $\alpha \circ\!\!\rightarrow \gamma$ as $\alpha \rightarrow \gamma$.*

- **R10:** *Suppose $\alpha \circ\!\!\rightarrow \gamma$, $\beta \rightarrow \gamma \leftarrow \theta$, $p_1$ is an uncovered p.d. path from $\alpha$ to $\beta$, and $p_2$ is an uncovered p.d. path from $\alpha$ to $\theta$. Let $\mu$ be the vertex adjacent to $\alpha$ on $p_1$ ($\mu$ could be $\beta$), and $\omega$ be the vertex adjacent to $\alpha$ on $p_2$ ($\omega$ could be $\theta$). If $\mu$ and $\omega$ are distinct, and are not adjacent, then orient $\alpha \circ\!\!\rightarrow \gamma$ as $\alpha \rightarrow \gamma$.*

### A.3 Soundness and completeness of Cluster-PC

**Theorem 4** (**Soundness and completeness of C-PC**)**.** *When the C-DAG $G_C$ is compatible with the ground truth DAG $G$, the C-PC algorithm as stated in Algorithm 2 is sound and complete (when using a CI oracle). Sound and complete in the sense that it returns the same MPDAG as using PC on $G$ and orienting additional edges according to $B$ (the pairwise causal constraint set induced by $G_C$).*

*Proof:* Let $\hat{G}$ be the output of the C-PC algorithm and $\bar{G}$ the MPDAG of the restricted Markov equivalence class $[G^*, B]$, obtained by restricting the CPDAG output from PC with $B$. $G$ is the ground truth DAG. $V$ is the node set and $E_{\hat{G}}, E_{\bar{G}}, E_{G^*}$ their edge sets, respectively. First, we show that $\hat{G}$ has the same skeleton as $\bar{G}$. Let $X - Y \in E_{\bar{G}}$ be any edge in $\bar{G}$. This means $X, Y$ are not d-separable in $G$, and thus any compatible C-DAG $G_C$ will not put $X, Y$ into non-adjacent clusters. Also, no CI test performed in C-PC will remove this edge

by the global Markov property ($\forall S : X \perp\!\!\!\perp_G Y | S \Rightarrow X \perp\!\!\!\perp Y | S$, thus $\forall S : X \not\perp\!\!\!\perp Y | S \rightarrow X \not\perp\!\!\!\perp_G Y | S$). On the other hand, let $X, Y$ be non-adjacent in $\bar{G}$, so $\exists S : X \perp\!\!\!\perp Y | S$. Without loss of generality, let $Y \notin de_X \cup nb_X^+$, then by Proposition 1 $mns_X(Y) \subset pa_X$ and $X \perp\!\!\!\perp Y | mns_X(Y)$. Furthermore, $mns_X(Y) \subset pa_X \subset nch_X$ and for every $S' \subset nch_x$, the CI test $X \overset{?}{\perp\!\!\!\perp} Y | S'$ is performed in C-PC. Thus the conditional independence $X \perp\!\!\!\perp Y | mns_X(Y)$ will be found and $X, Y$ are non-adjacent in $\hat{G}$, too. Second, we have to show that $\hat{G}$ and $\bar{G}$ have the same arrowheads. Due to the same skeleton, it is clear that they will also have the same unshielded colliders and same orientations due to Meek's edge orientation rules. The only thing left to show that a) additional edge orientations coming from the C-DAG edges $E_C$ are the same in $\hat{G}$ and $\bar{G}$, as well as that b) edge orientations from using Meek's orientation rules on the edges that partly come from a) are the same. Any directed edge $X \rightarrow Y \in E_{\bar{G}}$ coming from $B$ will also be directed in $\hat{G}$ due to Algorithm 1. This then also leads to any directed edge $X \rightarrow Y \in E_{\bar{G}}$ coming from using orientation rules when combining CPDAG $G^*$ with $B$, also being oriented in the last steps of C-PC when Meek's orientation rules are applied, so $X \rightarrow Y \in E_{\hat{G}}$. $\square$

### A.4 Soundness of C-FCI, informativeness vs FCITiers

**Theorem 5** (**Soundness of Cluster-FCI**). *If the C-DAG $G_C$ is compatible with ground truth MAG $G_M$, C-FCI is sound in the sense that an edge between any nodes $X_i, X_j$ is present in the PAG $G_P$ output from C-FCI if and only if it is present in the ground truth MAG $G_M$. Any arrow or tail edge mark in $G_P$ is also present in $G_M$.*

*Proof:* Cluster-FCI is the same as FCI, the only difference is it uses Algorithm 3 as an "oracle"pre-processing step. As C-DAG $G_C$ is compatible with $G_M$, and any nodes $X_i, X_j$ that are potentially connected by an inducing path are connected during Algorithm 3, using this Algorithm does not remove any edges that FCI would not also remove. With the same reasoning, any arrowhead present in the partial ancestral graph $G_P$ is also present in $G_M$.

**Remark 1** (**Sketch for proof: C-FCI at least as informative as FCITiers**). *Showing that C-FCI is at least as informative as FCITiers could be done as follows:*

(i) *Construct tiers from C-ADMG $G_C$ by combining clusters, so that the tiers are TBK. (Otherwise, one can not compare; FCITiers can only work on TBK, not on arbitrary C-ADMGs)*

(ii) *Show that any arrowhead oriented by FCITiers running on the previous TBK would also be oriented by running C-FCI on the C-DAG.*

*As the C-ADMG can contain bidirected edges, we can transform the C-ADMG to TBK as follows:*

(i) *Order clusters $C_1, \ldots, C_r$ topologically.*

(ii) *Group all clusters that are connected by a bidirected path together into one tier $T_i$.*

(iii) *The new cluster graph (now a C-DAG, without bidirected edges) could contain cycles. For any cycles in the C-DAG, merge the clusters on a cycle together into the same tier.*

(iv) *Sort the tiers topologically.*

(v) *Now one has a C-DAG that satisfies TBK.*

*C-FCI and FCITiers orient edge marks using the same collider rules and orientation rules. In addition, any extra edge marks in FCITiers come from two nodes $X, Y$ being in different tiers, e.g., $X \rightarrow Y$. But by construction, $X, Y$ were also in different clusters and if $X \rightarrow Y$ in TBK, then also $X \rightarrow Y$ in the partial mixed graph $G_{pm}$ from which C-FCI will start from. As again they use the same orientation rules, any arrow or tail edge mark from FCITiers will also be returned by C-FCI.*

## B    Supplement to the simulation studies

### B.1    Metrics for simulation studies

**Definition 24** (**Precision, recall, F1-score**). *The precision is defined as*

$$precision = \frac{TP}{TP + FP},\tag{2}$$

*where TP = true positives and FP = false positives. Positive means the corresponding edge is present and negative means it is absent.*

*The recall is defined as*

$$recall = \frac{TP}{TP + FN},\tag{3}$$

*where FN = false negative, i.e., an edge was erroneously deleted. The F1-score is the harmonic mean of recall and precision and encourages balance between the two, as it is zero whenever one of them is zero,*

$$F1\text{-}score = \frac{precision * recall}{precision + recall}.\tag{4}$$

**Definition 25** (**Structural Hamming distance**). *The structural Hamming distance (SHD) between graphs $G, G'$ is the number of edge deletions, additions or flips needed to transform $G$ into $G'$.*

**Remark 2** (**How arrow precision and recall are calculated**). *Adjacency true/false positive/negative is easy to understand. For arrow marks, positive/ negative refers to arrow edge marks, so positive means arrow is there, negative means arrow is not there (tail/ circle edge mark). For example, a false positive is there if the true MAG says there is a circle edge mark, but the algorithm output says there is an arrow edge mark (at some edge between some nodes).*

### B.2    Parameters and additional tables from the simulation studies

See Tables 4 to 6.

### B.3    How compatible C-DAGs are generated

We called the method we used for the simulation studies 1-3 "dag-first". In this case we first generate a DAG and afterwards create a clustering by slicing up the topological ordering into $n$ clusters of random size.

For example, if the DAG has ten nodes and the number of clusters is three, this method will select two numbers between $[1, n\_clusters]$, say four and ten. The first cluster will include the first three nodes in the topological ordering, the second cluster contains nodes four to nine and the third cluster contains node ten.

For Simulation 4 we use another method we call "cdag-first". This is because we wanted to enforce TBK on the cluster graphs. This method first generates an Erdős–Rényi graph for the clusters, for example of size three again. Then it generates nodes for each cluster so that they sum up to the desired node number, say ten again. Then the graph is built according to the generated cluster graph and nodes, and some edges from that graph are dropped out, that probability is influenced by the n_edges parameter. Since the C-DAG is generated first, we can exclude bidirected edges and a fully connected version of this C-DAG does satisfy TBK.

## C    Pairwise characterization of C-DAGs

A C-DAG can be represented as a boolean combination of pairwise background knowledge due to the following theorem:

**Theorem 6** (**Pairwise characterization of C-DAGs**). *Clusters $C_i, C_j$ imply, depending on their relationship in C-DAG $G_C$, pairwise constraints in the following ways:*

Table 4: Hyperparameters for simulation studies 1–4.

|  | Simulation 1 | Simulation 2 | Simulation 3 | Simulation 4 |
|---|---|---|---|---|
| Algorithms | PC, C-PC | PC, C-PC | FCI, FCITiers, C-FCI | FCI, FCITiers, C-FCI |
| Total number of graphs | 12600 | 3240 | 1620 | 1620 |
| Runs per configuration | 10 | 1 | 10 | 20 |
| DAG generation method | Erdős–Rényi | [Erdős–Rényi, hierarchical, scale-free] | Erdős–Rényi | Erdős–Rényi |
| Distribution | Gaussian | [Exponential, Gaussian, Gumbel, Logistic, MIM] | Gaussian | Gaussian |
| Alpha for CI test | [0.01, 0.05, 0.1, 0.25, 0.5] | 0.05 | 0.05 | 0.05 |
| CI test | fisherz | fisherz | fisherz | fisherz |
| Number of nodes | [15, 20, 30] | [15, 20, 25] | [18, 24, 36] | [15, 20, 30] |
| Number of edges | [15, 30, 50, 80] | [25, 30, 50, 80] | [22, 29, 36] | [15, 20, 25] |
| Number of clusters | [1, 2, 3, 4, 5, 6, 7] | [1, 2, 3, 4, 5, 6] | [2, 3, 4, 5, 6, 7] | [3, 4, 5] |
| Sample size | [300, 1000, 3000] | [300, 1000, 3000] | [300, 1000, 3000] | [300, 1000, 3000] |
| Weight range | (-1, 2) | (-1, 2) | (-1, 2) | (-1, 2) |
| Cluster method | dag-first | dag-first | dag-first | cdag-first |

| DAG method | Adj. precision (PC) | Adj. precision (C-PC) | Adj. recall (PC) | Adj. recall (C-PC) | Adj. F1-score (PC) | Adj. F1-score (C-PC) | SHD (PC) | SHD (C-PC) |
|---|---|---|---|---|---|---|---|---|
| Erdős–Rényi | 89.1% ± 0.5% | 86.4% ± 0.6% | 63.1% ± 1.3% | 67.7% ± 1.2% | 71.5% ± 1.0% | 74.1% ± 0.9% | 29.7 ± 1.3 | 24.9 ± 1.1 |
| Hierarchical | 92.0% ± 0.5% | 91.8% ± 0.5% | 26.2% ± 0.8% | 32.9% ± 0.8% | 39.4% ± 0.9% | 47.2% ± 0.9% | 116.1 ± 3.2 | 106.0 ± 3.1 |
| Scale free | 87.5% ± 0.6% | 84.8% ± 0.6% | 63.3% ± 1.1% | 68.2% ± 1.0% | 71.7% ± 0.9% | 74.2% ± 0.7% | 25.4 ± 0.9 | 21.4 ± 0.8 |

| DAG method | Arrow precision (PC) | Arrow precision (C-PC) | Arrow recall (PC) | Arrow recall (C-PC) | Arrow F1-score (PC) | Arrow F1-score (C-PC) |
|---|---|---|---|---|---|---|
| Erdős–Rényi | 58.4% ± 1.1% | 73.7% ± 0.9% | 34.1% ± 1.0% | 54.5% ± 1.3% | 41.6% ± 1.0% | 61.1% ± 1.1% |
| Hierarchical | 76.2% ± 1.0% | 85.3% ± 0.7% | 20.7% ± 0.8% | 30.1% ± 0.8% | 31.2% ± 1.0% | 43.2% ± 1.0% |
| Scale free | 58.4% ± 1.1% | 72.5% ± 1.0% | 34.3% ± 0.9% | 54.4% ± 1.2% | 41.8% ± 1.0% | 60.8% ± 1.1% |

Table 5: Simulation 2, varying graph generation method using gCastle (Zhang et al., 2021). The Base PC algorithm shows higher adjacency precision but generally lower recall and F1-score compared to Cluster-PC, a trend similar to Simulation 1. C-PC has better arrow metrics. The structural Hamming distance (SHD) reflects similar trends, with performance depending on the underlying DAG structure.

| Distribution | Adj. precision (PC) | Adj. precision (C-PC) | Adj. recall (PC) | Adj. recall (C-PC) | Adj. F1-score (PC) | Adj. F1-score (C-PC) | SHD (PC) | SHD (C-PC) |
|---|---|---|---|---|---|---|---|---|
| Exponential | 88.8% ± 0.6% | 86.7% ± 0.6% | 44.5% ± 2.0% | 51.2% ± 1.9% | 55.2% ± 1.8% | 60.8% ± 1.6% | 60.5 ± 4.5 | 54.5 ± 4.3 |
| Gaussian | 88.6% ± 0.6% | 86.6% ± 0.6% | 44.5% ± 2.0% | 51.6% ± 1.9% | 55.2% ± 1.8% | 61.2% ± 1.6% | 60.7 ± 4.6 | 54.5 ± 4.2 |
| Gumbel | 89.6% ± 0.6% | 87.6% ± 0.6% | 45.1% ± 2.0% | 51.9% ± 1.9% | 55.9% ± 1.9% | 61.6% ± 1.6% | 60.9 ± 4.7 | 54.9 ± 4.4 |
| Logistic | 92.3% ± 0.8% | 90.8% ± 0.8% | 57.6% ± 1.5% | 60.7% ± 1.4% | 68.3% ± 1.1% | 70.4% ± 1.0% | 50.9 ± 3.3 | 44.1 ± 3.1 |
| MIM | 88.4% ± 0.9% | 86.6% ± 1.0% | 62.5% ± 1.7% | 65.9% ± 1.6% | 69.8% ± 1.2% | 71.6% ± 1.1% | 52.2 ± 3.6 | 45.7 ± 3.4 |

| Distribution | Arrow precision (PC) | Arrow precision (C-PC) | Arrow recall (PC) | Arrow recall (C-PC) | Arrow F1-score (PC) | Arrow F1-score (C-PC) |
|---|---|---|---|---|---|---|
| Exponential | 65.6% ± 1.4% | 78.1% ± 1.0% | 25.5% ± 1.3% | 43.0% ± 1.8% | 34.3% ± 1.5% | 52.4% ± 1.7% |
| Gaussian | 66.6% ± 1.3% | 77.6% ± 1.1% | 26.1% ± 1.3% | 43.0% ± 1.8% | 35.2% ± 1.4% | 52.4% ± 1.6% |
| Gumbel | 66.3% ± 1.4% | 78.3% ± 1.1% | 26.0% ± 1.4% | 43.1% ± 1.8% | 35.1% ± 1.5% | 52.6% ± 1.7% |
| Logistic | 62.3% ± 1.9% | 77.4% ± 1.5% | 33.5% ± 0.9% | 49.0% ± 1.3% | 41.9% ± 1.1% | 58.0% ± 1.2% |
| MIM | 61.0% ± 1.5% | 74.4% ± 1.3% | 37.6% ± 1.1% | 53.6% ± 1.5% | 44.5% ± 1.0% | 59.7% ± 1.2% |

Table 6: Simulation 2, varying distribution method using gCastle (Zhang et al., 2021). The Base PC algorithm shows higher adjacency precision but generally lower recall and F1-score compared to Cluster-PC, a trend similar to Simulation 1. C-PC has better arrow metrics. The structural Hamming distance (SHD) reflects similar trends, with performance depending on the underlying DAG structure.

*(i)*

$$C_i = C_j \Rightarrow \boldsymbol{T}$$

*(the tautology, which is always true and places no restriction)*

*(ii)*

$$C_i \quad \rightarrow \quad C_j \quad \Rightarrow \quad \bigwedge_{X_i \in C_i, X_j \in C_j} X_i \not\leftarrow\!\!\!- X_j \quad \wedge \bigvee_{X_i \in C_i, X_j \in C_j} X_i \quad \rightarrow \quad X_j \quad =: \quad dir(C_i, C_j) \quad (5)$$

*(note that $X_i \not\leftarrow\!\!\!- X_j$ also implies $X_i \not\leftarrow X_j$)*

*(iii)*

$$C_i \quad \dashrightarrow \quad C_j \wedge C_i \quad \not\rightarrow \quad C_j \quad \Rightarrow \quad \bigwedge_{X_i \in C_i, X_j \in C_j} X_i \not\leftarrow\!\!\!- X_j \wedge X_i \quad \not\rightarrow \quad X_j \quad =: \quad anc(C_i, C_j) \quad (6)$$

*(iv)*

$$C_i \not\dashrightarrow C_j \wedge C_i \not\leftarrow\!\!\!- C_j \wedge C_i \quad \neq \quad C_j \quad \Rightarrow \quad \bigwedge_{X_i \in C_i, X_j \in C_j} X_i \not\leftarrow\!\!\!- X_j \wedge X_i \not\dashrightarrow X_j \quad =: \quad nrel(C_i, C_j) \quad (7)$$

*These four points exhaust all possibilities in which two clusters could relate to each other in a C-DAG. The background knowledge implied by C-DAG $G_C$ with clustering $C = \{C_1, \ldots, C_m\}$ can therefore be represented as a boolean combination in pairwise form:*

$$bk(G_C) \quad := \bigwedge_{C_i, C_j \in C} ( \bigwedge_{C_i \rightarrow C_j} dir(C_i, C_j) \quad \wedge \bigwedge_{\substack{C_i \dashrightarrow C_j \\ C_i \not\rightarrow C_j}} anc(C_i, C_j) \quad \wedge \bigwedge_{\substack{C_i \neq C_j, C_i \not\dashrightarrow C_j \\ C_i \not\leftarrow\!\!\!- C_j}} nrel(C_i, C_j)). \quad (8)$$

*Thus the following equivalence holds:*

$$G \text{ compatible with } G_C \iff bk(G_C) \text{ is true for } G. \quad (9)$$

*Proof:* (i): For two nodes in the same cluster $C_i$, no restriction is made, so for all $X_i, X_j \in C_i$, $\mathbf{T}$ is true.

(ii): For two clusters $C_i, C_j$ with $C_i \rightarrow C_j$ and nodes $X_i \in C_i, X_j \in C_j$, it is impossible to have $X_i \leftarrow\!\!\!- X_j$, as that would mean $C_i \leftarrow\!\!\!- C_j$ in $G_C$, which is a cycle together with $C_i \rightarrow C_j$ in contradiction to $G_C$ being a C-DAG and $C$ being admissible. In addition, at least one $X_i \rightarrow X_j$ needs to be present by C-DAG construction. Therefore $C_i \rightarrow C_j$ implies $\bigwedge_{X_i \in C_i, X_j \in C_j} X_i \not\leftarrow\!\!\!- X_j \wedge \bigvee_{X_i \in C_i, X_j \in C_j} X_i \rightarrow X_j$.

(iii): Let two clusters $C_i, C_j$ have $C_i \dashrightarrow C_j \wedge C_i \not\rightarrow C_j$, i.e., there is a directed path from one to another, but they are not adjacent. With nodes $X_i \in C_i, X_j \in C_j$ for analogous reasoning to (ii), it is impossible to have $X_i \leftarrow\!\!\!- X_j$. In addition, as $C_i \not\rightarrow C_j$, it is also impossible to have $X_i \rightarrow X_j$. So $C_i \dashrightarrow C_j \wedge C_i \not\rightarrow C_j$ implies $X_i \not\leftarrow\!\!\!- X_j \wedge X_i \not\leftarrow X_j$.

(iv): For two clusters that are not the same, not adjacent and not connected by a directed path, i.e., $C_i \not\dashrightarrow C_j \wedge C_i \not\leftarrow\!\!\!- C_j \wedge C_i \neq C_j$ and nodes $X_i \in C_i, X_j \in C_j$ it is impossible for $X_i, X_j$ to be connected by a directed path. Therefore this implies $\bigwedge_{X_i \in C_i, X_j \in C_j} X_i \not\leftarrow\!\!\!- X_j \wedge X_i \not\dashrightarrow X_j$.

Equivalence statement: "$\Rightarrow$": Let $G = (V, E)$ be a graph over the same variables $V$ with edges $E$ being compatible with C-DAG $G_C$. Furthermore let $bk(G_C)$ be as defined in Eq. (8). The task is to show that $bk(G_C)$ is true for $G$, specifically, that any edge between any $X_i, X_j$ satisfies $bk(G_C)$. Take any $X_i, X_j \in V$ and their respective clusters $X_i \in C_i, X_j \in C_j$. $C_i, C_j$ relate to each other in exactly one of the four ways described in (i)-(iv). Whatever the cluster relationship is on the left hand side of (i)-(iv), the proofs of (i)-(iv) above show that the edge between $X_i, X_j$ satisfies the constraint on the right hand side of (i)-(iv). So the boolean pairwise combination $bk(G_C)$ is true for $G$.

"$\Leftarrow$": Let $G_C$ be a C-DAG and $bk(G_C)$ be true for DAG $G$. The task is to show that $G$ is compatible with $G_C$. $G = (V, E)$ is compatible with $G_C$, if none of its edges $E$ contradict the C-DAG edges $E_C$. Take any $X_i, X_j$ and their corresponding clusters $X_i \in C_i, X_j \in C_j$. The edge between $X_i, X_j$ can take any of the three forms $X_i \to X_j$, $X_i \leftarrow X_j$ or $X_i \not{\;} X_j$. Without loss of generality (no need to consider the case $C_i \leftarrow C_j$ due to symmetry), the C-DAG restriction on the edge between $X_i, X_j$ can take any of the forms in (i)-(iv).

If restriction $\mathbf{T}$ is put on the edge between $X_i, X_j$, they are put in the same cluster and any of $X_i \to X_j$, $X_i \leftarrow X_j$ and $X_i \not{\;} X_j$ would be compatible with $G_C$. If restriction $X_i \leftarrow\!\!\leftarrow X_j$ is put, $X_i$ and $X_j$ are in adjacent clusters $C_i \to C_j$. The edges allowed by $X_i \leftarrow\!\!\leftarrow X_j$ are $X_i \to X_j$ and $X_i \not{\;} X_j$. Both of them are compatible with $G_C$. If restriction $X_i \leftarrow\!\!\leftarrow X_j \wedge X_i \not\to X_j$ or $X_i \leftarrow\!\!\leftarrow X_j \wedge X_i \to X_j$ is put, only $X_i \not{\;} X_j$ is allowed and $C_i, C_j$ are not adjacent, so $X_i \not{\;} X_j$ is compatible with $G_C$. $\square$

**Theorem 7 (Pairwise characterization of C-ADMGs).** *Let the setup be the same as in Theorem 6. To include bidirected constraints, the rules (i)-(iv) from Theorem 6 get extended by*

*(v)*

$$C_i \not\leftrightarrow C_j \iff \bigwedge_{X_i \in C_i, X_j \in C_j} X_i \not\leftrightarrow X_j =: nlat(C_i, C_j)$$

*The background knowledge is then denoted as*

$$bk(G_C) = \bigwedge_{C_i, C_j} \left( \bigwedge_{C_i \to C_j} dir(C_i, C_j) \wedge \bigwedge_{\substack{C_i \dashrightarrow C_j \\ C_i \not\to C_j}} anc(C_i, C_j) \right.$$

$$\left. \wedge \bigwedge_{\substack{C_i \not\!\!\to C_j \\ C_i \not\!\!\leftarrow C_j}} nrel(C_i, C_j) \wedge \bigwedge_{C_i \not\leftrightarrow C_j} nlat(C_i, C_j) \right) \quad (10)$$

*and*

$$G \text{ compatible with } G_C \iff bk(G_C) \text{ is true in } G \quad (11)$$

*Proof:* (v) If $C_i \not\leftrightarrow C_j$, by C-ADMG definition that means for all $X_i \in C_i, X_j \in C_j$ it is $X_i \not\leftrightarrow X_j$ which exactly implies $nlat(C_i, C_j)$. For the reverse direction, if $nlat(C_i, C_j)$ it means for no $X_i \in C_i, X_j \in C_j$ it is $X_i \leftrightarrow X_j$. This means the C-ADMG does not have $C_i \leftrightarrow C_j$. The rest follows analogously to the proof of Theorem 6. $\square$

# D    Using C-DAGs for score-based and continuous optimization discovery algorithms

In this paper we studied constraint based causal discovery, but DAGs can also be estimated via a constrained optimization problem (Chickering, 2002; Zheng et al., 2018). Such a problem typically admits a form like

$$\min_{G \in \mathbb{R}^{d \times d}} F(G) \quad \text{subject to} \quad G \in \text{DAGs}$$

with $F(G)$ evaluating the goodness of fit of a graph $G$ to the available data. One can easily extend this problem to include C-DAG constraints:

$$\min_{G \in \mathbb{R}^{d \times d}} F(G) \quad \text{subject to} \quad G \in \text{DAGs}, G \text{ compatible with } G_C$$

so that the optimization procedure is only able to take steps in the space of DAGs that are compatible with C-DAG $G_C$ or only able to return optimal solutions that are compatible with $G_C$. This could be an interesting topic for future research.

# E    Application domains of C-DAG based causal discovery

In this section we provide four examples that motivate why C-DAG based causal discovery is practically relevant. The key advantage of C-DAGs over tiered background knowledge (TBK) is their ability to represent

v-structures at the cluster level ($C_1 \rightarrow C_2 \leftarrow C_3$), where two clusters independently cause a third but are not causally related to each other. Such structures are ubiquitous in real-world systems but impossible to encode with TBK, which requires a total ordering of tiers. This makes C-DAGs strictly more expressive than TBKs as shown in Section 2.1. When working with high-dimensional data, clustering variables into meaningful groups is a natural and widely-adopted strategy to manage complexity. As Kelly et al. (2022) note in their review of causal discovery for molecular networks: "Many studies have identified smaller subsets of genes through previous knowledge of pathways or clustering of undirected networks before inferring causality." C-DAGs formalize this common practice while preserving the flexibility to encode non-hierarchical causal relationships between clusters.

**Gene Regulatory Networks and Multi-Omics Integration.** Gene Regulatory Networks and Multi-Omics Integration Genes naturally cluster by functional pathways (Reactome, Gene Ontology). V-structures are common: for example, the p53 pathway and Wnt pathway both independently regulate cell cycle progression, forming a v-structure that TBK cannot represent. Dugourd et al. (2021a) demonstrate causal integration across signaling, transcription, and metabolism using prior knowledge networks, exactly the setting where C-DAGs apply.

**Climate Science**. Climate variables cluster by physical processes (ocean, atmosphere, cryosphere) and spatial regions. Runge et al. (2019b) developed PCMCI for climate causal discovery, demonstrating that the Walker Circulation can be inferred from grouped climate indices. V-structures examples: ENSO independently affects both Indian monsoons and African rainfall patterns, while these regional effects are not causally related to each other. Additionally, ocean-atmosphere coupling involves bidirected relationships—requiring C-ADMGs. Nowack et al. (2020) use causal networks for climate model evaluation, grouping variables by physical domain.

**Social Determinants of Health (SDOH).** The epidemiological "web of causation" MacMahon (1960) explicitly rejects linear causal chains. V-structures occur often: socioeconomic factors and genetic predisposition independently affect disease risk. Korvink et al. (2025b) apply causal discovery with dimension reduction to map SDOH relationships, noting that "For meaningful analysis, clusters of related variables within an SDOH domain must be compressed into a single latent construct", making C-DAGs a natural choice.

**Macroeconomics.** Economic variables cluster by sector (monetary, fiscal, real economy, external). Nakamura & Steinsson (2018) discuss how identification in macroeconomics requires structural assumptions about contemporaneous relationships. Example V-structures arising in policy transmission: monetary and fiscal policy independently affect output through different channels.

# F    Comparison of Algorithms on Sachs Protein Dataset

We use the Sachs dataset on protein signaling networks from Sachs et al. (2005), provided by the causal-discovery-toolbox (Kalainathan et al., 2020) python package. The ground truth graph (no DAG, contains one cycle) is shown in Fig. 7. To ensure robustness of our analysis, we bootstrap (with replacement) 100 datasets and run PC, k-PC (with $k = 2$) and Cluster-PC discovery 100 times each. We use the Fisher-z CI test and $\alpha = 0.01$. We assume expert knowledge gives access to the C-DAG shown in Fig. 8a. The graphs estimated on the true (no bootstrapped data) are shown in Figs. 8b to 8d.

The results of our experiment are shown in Table 7. The performance of all algorithms w.r.t. adjacency metrics is very close, while k-PC has low arrow precision but high arrow recall compared to PC and Cluster-PC. Cluster-PC has the lowest SHD, PC being a close second and k-PC farther away.

Table 7: Comparison of PC, k-PC ($k = 2$) and Cluster-PC for 100 bootstrapped Sachs datasets.

| Method | Adj. precision | | Adj. recall | | Adj. F1-score | | Arrow precision | | Arrow recall | | Arrow F1-score | | SHD |
|---|---|---|---|---|---|---|---|---|---|---|---|---|---|
| PC | 45.0% 0.7% | ± | 56.9% 0.7% | ± | 50.2% 0.6% | ± | 31.3% 1.7% | ± | 37.4% 2.3% | ± | 34.0% 1.9% | ± | $22.9 \pm 0.4$ |
| k-PC | 42.3% 0.6% | ± | 63.0% 1.0% | ± | 50.6% 0.7% | ± | 22.5% 0.5% | ± | 55.9% 1.4% | ± | 32.1% 0.7% | ± | $30.9 \pm 0.3$ |
| Cluster-PC | 43.7% 0.6% | ± | 55.7% 1.0% | ± | 49.0% 0.7% | ± | 32.6% 1.0% | ± | 35.8% 1.2% | ± | 34.1% 1.1% | ± | $21.5 \pm 0.3$ |

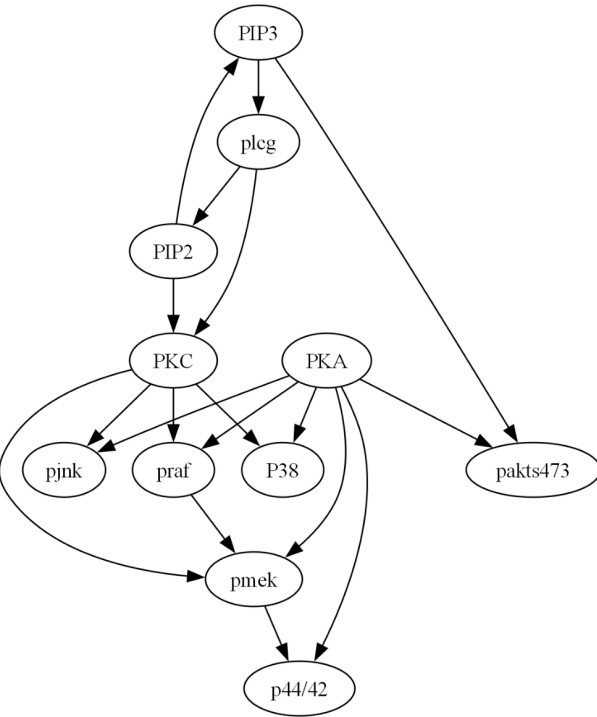

Figure 7: Ground truth graph (no DAG, contains one cycle) of the Sachs protein signaling dataset from the causal-discovery-toolbox.

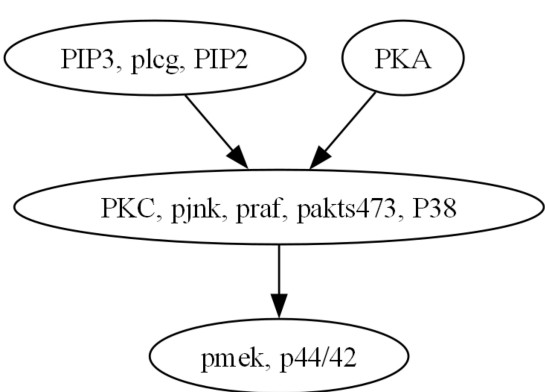

(a) C-DAG we used for Cluster-PC on the Sachs dataset.

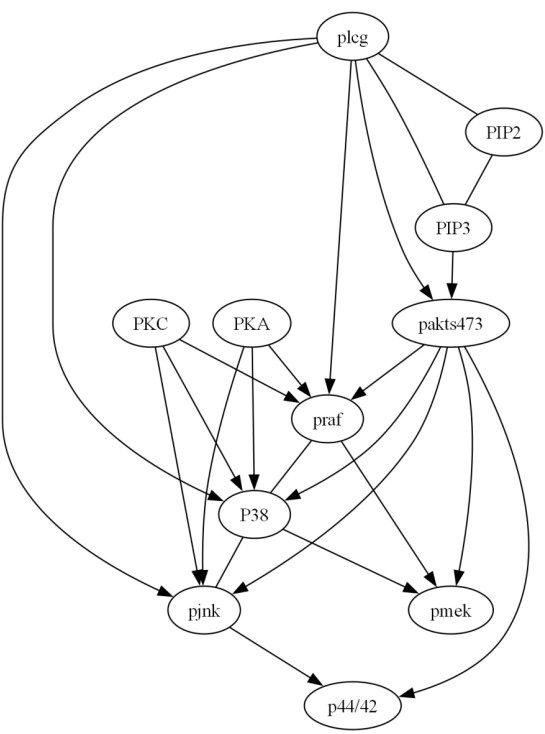

(b) Cluster-PC estimated graph based on the true (no bootstrap) Sachs dataset.

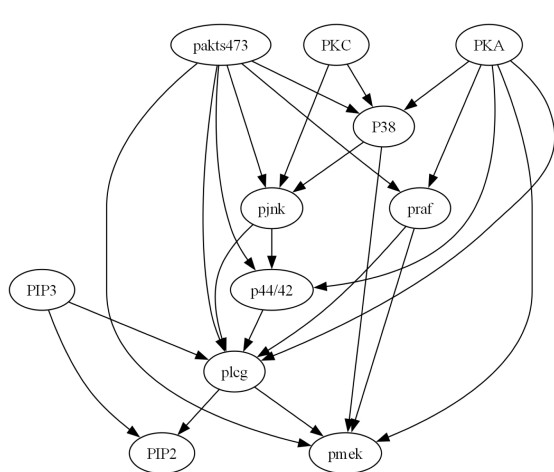

(c) PC estimated graph based on the true (no bootstrap) Sachs dataset.

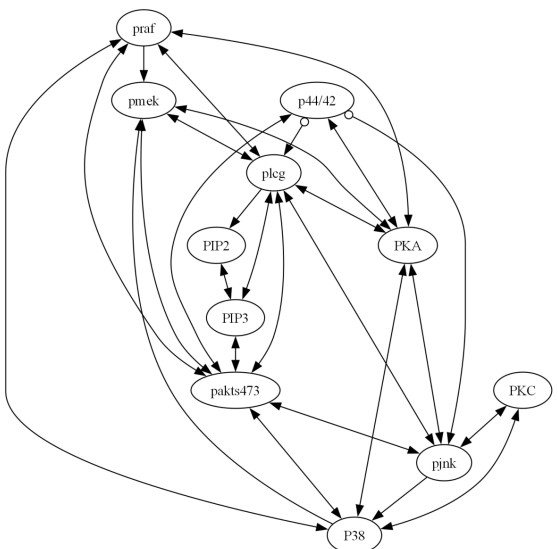

(d) k-PC ($k = 2$) estimated graph based on the true (no bootstrap) Sachs dataset.

