# OpenReview forum: "Cluster-Dags as Powerful Background Knowledge For Causal Discovery"
_TMLR — Accepted by TMLR_

### Review · Reviewer_DjRy · 2026-02-22

**Summary Of Contributions:**

The paper formalizes how to use C-DAGs and C-ADMG as an initial solution for the PC and the FCI algorithm. In practice. Cluster-graphs imply certain conditional independencies (as was proven for C-DAGs by Anand et al., 2023), so that when running constraint-based causal discovery less tests are needed (given that some independence relations are already encoded in the cluster-graph).

The paper is generally well written. I am, however, a bit skeptical about the interest of the contributions. There is no clear motivation for why cluster DAGs are an interesting setting to consider, meaning, to me, it is unclear why having access to cluster DAGs would be common in practical scenarios. On top of that, I find that the novel contributions in this work are quite marginal. In details:

1. The soundness and completeness of PC on top of a C-DAG is a rather straightforward corollary of Theorem 1 from Anand et al., 2023. In fact, the C-DAG naturally acts as a pre-pruning of edges, and the PC algorithm (or any other method of preference, like, e.g., LiNGAM if one were to make parametric assumptions) can be used to prune the remaining ones.
2. The only novel and meaningful intuition, in my opinion, is that C-ADMGs are not the right starting point for running FCI, and so there is some preprocessing to ensure some kind of maximality before running the FCI algorithm.

All in all, I judge that the novelty of this paper is marginal; however, the paper is well-written and could be interesting to a part of the community, especially the section on the use of C-ADMG. In this regard, to recommend acceptance, it is necessary to add convincing arguments that Cluster DAGs are a 1. common 2. desirable way to encode background knowledge: can you cite real-world scenarios in which this is the case?

**Audience:**

Yes

**Audience Explanation:**

This is the part I am more skeptical about. In fact, as I write in the "summary of the contributions", I find the contributions somewhat marginal. However, the authors identify the right way to use C-ADMG as a warm start for FCI and prove the soundness of the resulting Cluster-FCI algorithm: if they can provide convincing cases in which their algorithmic procedure would be useful (i.e., real-world cases in which C-ADMGs and C-DAGs are accessible), then the content of this paper can be valuable for practitioners.

**Claims And Evidence:**

Yes

**Claims Explanation:**

The technical contribution is solid and clean, as well as the writing. I praise the author for their clarity and for correctly discussing prior work that is fundamental to theirs (Anand et al., 2023)

**Requested Changes:**

I would like to see real-world use cases in which C-DAGs and C-ADMGs are accessible; this way, I would be convinced that the algorithmic contributions of this work have a practical value.

---

### Review · Reviewer_9ZM2 · 2026-02-25

**Summary Of Contributions:**

- Authors work introduces C-DAGs, a new way to represent background knowledge in causal discovery. C-DAGs are more flexible than previous approaches, like tiered background knowledge (TBK).
- Authors introduce Cluster-PC and Cluster-FCI, adaptations of PC and FCI that leverage C-DAGs. They prove that Cluster-PC is both sound and complete, while Cluster-FCI is sound but not guaranteed to be complete by design.
- Simulations show that these methods outperform standard algorithms that don’t use background knowledge.

**Audience:**

Yes

**Audience Explanation:**

The paper has enough theoretical novelty and methodological contribution that members of TMLR's audience, especially those interested in causal discovery, would find it worth knowing about, even if the empirical results are preliminary.

**Claims And Evidence:**

No

**Claims Explanation:**

- The formalization of C-DAG constraints are well-supported.
- However, the simulation evidence supporting the empirical effectiveness of Cluster-PC and Cluster-FCI is limited:
    - The experiments use a fixed number of nodes and a fixed number of samples, which restricts the generalizability of the results to larger or smaller graphs and varying data regimes.
    - Additionally, all simulations are performed with continuous variables, leaving open the question of how these methods perform on discrete (e.g. categorical) or mixed-type distributions, which are common in real-world applications.
    - While authors correctly state that: "current methods face several challenges, especially when dealing with high-dimensional data and complex dependencies", this narrow experimental design makes it difficult to fully assess the usefulness of the proposed methodology.
    - While the simulations indicate that the proposed methods outperform non-background-knowledge baselines in the specific scenarios tested, they do not yet provide conclusive evidence that these gains will persist under more diverse or realistic conditions.

**Requested Changes:**

- Expand the simulations by varying distribution type, number of nodes, sample size. Please, note that varying the number of edges means that the associated distribution has more parameters, hence, you need to adjust the sample size accordingly.
- Since you appear to perform simulations on batch of graphs, with multiple runs even, please, do report the confidence intervals of the evaluation metrics, so that we can assess the significance of the metrics variance.

---

### Review · Reviewer_E4Gp · 2026-02-27

**Summary Of Contributions:**

This paper modifies standard constraint-based causal discovery algorithms with respect to background knowledge provided as clustered nodes. Cluster-DAG formulation is borrowed from Anand et al. (2023) and organizes nodes into clusters in which known causal relations dictate how clusters are formed. Authors argue that such groupwise background knowledge for causal discovery is often available yet is an underexploited research direction, which I find reasonable.

The paper shows that Cluster-DAG is a strictly more flexible way to encode background knowledge than some other common approaches (TBK - tier-based knowledge), modifies standard PC/FCI algorithms to use the provided background cluster knowledge, and significantly reduces the number of required CI tests. The proposed Cluster-PC modification is shown to be sound and complete, which is somewhat a straightforward result. Finally,  the paper presents simulations on synthetic data to show that the reduced number of CI tests leads to better accuracy as the number of clusters increases.

On the positive side, all the above points are valid and reasonable. However, they are also very incremental, and I’m not sure the evidence is strong enough to convince the reader to change how to encode/use the background knowledge in causal discovery. This judgement is due to the motivation of the paper not being well-supported via examples, the performance gain being modest, and the lack of demonstration on a potential real use case. I expand on these comments in the following boxes.

**Additional Comments:**

**Continuous optimization-based causal discovery:** Is comparison to NOTEARS-type of work possible? Appendix D suggests that it should be possible (correct me if I misinterpret). I understand that the focus is on constraint-based algorithms, but from another perspective, if constraint-based methods are unreliable in a given setting, then minor improvements in that setting wouldn’t move the needle to make them more useful (e.g., going from 26.3% F1 score to 28.6% almost doesn’t mean anything to a practitioner, in my subjective opinion).

**TBK condition:** Is it possible to stratify the generated graphs into two groups according to satisfying TBK or not? I know that Simulation 4 aims to demonstrate that, but it only shows one way of generating graphs that satisfy TBK. On the other hand, I suppose that some graphs generated without restrictions (Simulation 3) would also satisfy TBK, and it’d be interesting to see the difference between C-FCI and FCITiers closer.

**Bounded-size CI tests:** This is not a request but just a suggestion, as it might not be exactly relevant. There is recent work on $k$-PC (Kocaoglu, 2023) that investigates PC algorithm when the size of the conditioning set is upper bounded by $k$. In some sense, C-DAG gives you similar information — not directly bounding the conditioning set but effectively telling you which nodes you don’t need to bother to include in potential separating sets, i.e., decreasing the potential maximum size of the conditioning sets. Is it possible to compare the number of CI tests and the performance of Cluster-PC with k-PC? This could be an interesting study, since the performance of Cluster-PC is not perfect anyway, e.g., the trade-off between reducing the number of CI tests via bounding k vs. using cluster knowledge.

Some minor comments that don’t affect my judgement of the paper:
- I suggest that the authors categorize the related work section. The current section provides an exhaustive coverage, which I appreciate. However, it’s a bit unstructured, reducing its usefulness and making the reading more difficult.
- I don’t think causal representation learning / causal disentanglement needs to be discussed in the related work, as it is a very different problem, but if the authors think it’s good to have it (I suppose so as there is a reference to Li et al. (2024)), then Morioka and Hyvärinen (2024) is perhaps the most related one.
- Minor nitpicking: There are some ‘[word]’ usage, e.g., ‘oracle’ in Theorem 5 proof, it could be good to fix it in the revision.

Hiroshi Morioka and Aapo Hyvärinen. "Causal representation learning made identifiable by grouping of observational variables." ICML 2024.

Murat Kocaoglu. "Characterization and learning of causal graphs with small conditioning sets." NeurIPS 2023.

**Audience:**

Yes

**Audience Explanation:**

Yes. TMLR’s audience includes a large causal discovery community, which could be interested in the findings of this paper — even though I find the results be incremental, formalization is still valuable, and with improved evidence, it could be a useful paper to the community.

**Broader Impact Concerns:**

The work does not require adding a broader impact statement.

**Claims And Evidence:**

No

**Claims Explanation:**

The claims made in the paper are all reasonable and accurate to my judgment. However, they are not strongly convincing, and the evidence is not too strong either. That being said, I think these points could be addressable, see my requested changes below.

**Requested Changes:**

**[Major] Practical importance:** One focus of the paper is that:

>C-DAGs are strictly more flexible than TBK (tier-based knowledge)

It is true, but I think it’s important to provide some practical examples, to make the case more convincing. For instance, the paper argues that groupwise background knowledge is often available. Yet, the only examples seem to be two vague references to Riberio et al. (2025) and  Anand & Hripcsak (2025) in page 3, without discussing why such applications require the use of C-DAGs (i.e., why TBK doesn’t apply).

**[Medium]** Related to the above major point: It’d be much more interesting to apply Cluster-PC/FCI to some real data and demonstrate the benefits compared to standard PC and/or FCITiers. In general, I value the algorithmic advances if the benefit is undeniably strong, even if it’s only shown on synthetic data, but given that the algorithmic improvement is rather incremental, I’d have liked to see a stronger experimental demonstration. Overall, results are as expected -- if you reduce the number of CI tests via prior knowledge, then you can expect that the performance will improve. Though the improvement also seems not too dramatic, that’s why I think showing some *real* benefit would be great.

**[Medium]**: According to Table 3, all studies use a fixed number of nodes. Seeing the effect of the number of nodes would have been better.

**[Medium/minor]** The paper says
>Finally, we conjecture that for ancestral C-ADMGs, C-FCI is also complete, i.e., all counterexamples have to rely on non-ancestral C-ADMGs as background knowledge. This also remains an open question for future work.

I think this is true (or say, I would be surprised if this is not true). Can authors comment on their effort in addressing this question? I’m asking it because by itself, it would be an incremental result in future work, but it can make the findings of this paper strictly more interesting.

**[Minor]**
>Further exploring this deviation from the well-explored setting of relying primarily on ancestral graphs for causal discovery is an interesting direction for future work.

Similarly, to a lesser extent, exploring this setting would have been welcomed in the present paper.

**[Minor]** Algorithm 2 Cluster-PC algorithm: I’d have liked to see a (brief) discussion on the difference from the original PC. If I’m reading correctly, it is a minor modification of PC, and Theorem 2 (soundness and completeness of Cluster-PC) follows straightforwardly. Still, I think it is useful to have a formalized algorithm, but its development and benefits could’ve been made clearer.

---

### Decision · Action_Editor_b8cT · 2026-04-28

**Recommendation:** Accept as is

**Audience:**

Yes

**Audience Explanation:**

The paper would be of interest to researchers working on causal discovery.

**Claims And Evidence:**

Yes

**Claims Explanation:**

The paper investigate the idea of incorporating Cluster-DAGs as a prior knowledge for warm-start causal discovery. The idea is not totally novel, but the authors did a good job at providing evidence supporting the claims.

---

> ### Author Response · Authors · 2026-05-18
> **Camera ready version submitted**
>
> Dear AE, thank you for your response! We have submitted the camera ready version.